# Olfactory receptor neurons use gain control and complementary kinetics to encode intermittent odorant stimuli

**Srinivas Gorur-Shandilya[1,2†], Mahmut Demir[2†], Junjiajia Long[2,3], Damon A Clark[1,2,3*], Thierry Emonet[1,2,3*]**

[1]Interdepartmental Neuroscience Program, Yale University, New Haven, United States; [2]Department of Molecular, Cellular, and Developmental Biology, Yale University, New Haven, United States; [3]Department of Physics, Yale University, New Haven, United States

**Abstract** Insects find food and mates by navigating odorant plumes that can be highly intermittent, with intensities and durations that vary rapidly over orders of magnitude. Much is known about olfactory responses to pulses and steps, but it remains unclear how olfactory receptor neurons (ORNs) detect the intensity and timing of natural stimuli, where the absence of scale in the signal makes detection a formidable olfactory task. By stimulating *Drosophila* ORNs in vivo with naturalistic and Gaussian stimuli, we show that ORNs adapt to stimulus mean and variance, and that adaptation and saturation contribute to naturalistic sensing. Mean-dependent gain control followed the Weber-Fechner relation and occurred primarily at odor transduction, while variance-dependent gain control occurred at both transduction and spiking. Transduction and spike generation possessed complementary kinetic properties, that together preserved the timing of odorant encounters in ORN spiking, regardless of intensity. Such scale-invariance could be critical during odor plume navigation.

*For correspondence: damon. clark@yale.edu (DAC); thierry. emonet@yale.edu (TE)

†These authors contributed equally to this work

**Competing interests:** The authors declare that no competing interests exist.

## Introduction

Insects navigate odor landscapes that are often not smooth gradients (*Cardé and Willis, 2008*; *Riffell et al., 2008*). Instead, turbulent airflows shape odor plumes into intermittent whiffs separated by stochastic durations of background air (blanks). In the absence of reliable spatial gradients, navigating insects may combine the timing of whiff encounters (*Vergassola et al., 2007*) with sensation of wind direction (*Budick and Dickinson, 2006*; *Cardé and Willis, 2008*; *Duistermars et al., 2009*) to navigate odor plumes towards mates and food. Insect olfactory systems face dual challenges in detecting natural odor plumes. First, the intensity of whiffs is typically distributed according to a power law (*Murlis et al., 1992*), with intense whiffs interleaved unpredictably with weak ones (*Riffell et al., 2008*). Second, whiff durations and blank durations are also distributed as a power law over a wide range of time scales (*Celani et al., 2014*). The encoding problem is aggravated by shifting local statistics of odor encounters, which change with wind speed (*Nagel and Wilson, 2016*, *2011*), position (*Justus et al., 2002*), or environment (*Murlis et al., 2000*). How does the olfactory system manage to encode whiffs of odors whose intensities and timing can vary over such wide ranges?

Several features of the olfactory system contribute to encoding odor stimuli of different intensities. A single odorant can be detected by multiple receptor types, with different sensitivities (*Hallem and Carlson, 2006*). Static compressive nonlinearities at both olfactory receptor neurons (ORNs) and their post-synaptic targets, the projection neurons (PNs), selectively amplify weak signals

**eLife digest** Insects follow odor trails carried by the wind to find mates and sources of food. The turbulent motion of the air means that these odors tend to arrive in whiffs with varying intensities and durations, which makes it difficult to distinguish them. Insects use sensory cells called olfactory receptor neurons on their antenna to process odors. Specialized receptor proteins on the surface of these olfactory receptor neurons detect odor molecules and set off a cascade of events in these cells that ends with a signal being sent to the brain.

Much is known about how insects detect and process different kinds of smells, but it remains less clear how their olfactory neurons process the timing and intensity of odor whiffs. Now, Gorur-Shandilya et al. report what happens in the olfactory receptor neurons of fruit flies when they have to compensate for variations in the duration and intensity of odor whiffs.

In the experiments, fruit flies were exposed to two sweet-smelling odors. To do so, Gorur-Shandilya et al. built an apparatus that enabled them to control the airflow with enough precision that they could simulate the variability in the timing and intensity of natural odors in the air. The response of the flies' olfactory receptor neurons to these smells was recorded. The experiments showed that the neurons could adapt to both the average intensity and the variance in intensity of odor signals.

The ability of these neurons to adapt to the average intensity of the odors followed a specific pattern, which is also seen in sensory cells responsible for vision and touch. Adapting to the average strength of an odor slows down the first of two steps in its processing. However, the second step has a complementary mechanism to speed up signals to the brain, so the timing of an odor whiff is accurately captured regardless of how strong it is. Based on these results, Gorur-Shandilya et al. created a biophysical model that could reproduce the experimental data, including the slowdown in the first step.

The experiments and the model may now help other scientists to investigate how different animals detect and process smells. For example, some insects are pests of agricultural crops, while other insects, such as mosquitos, spread diseases between people. A better understanding of how insects detect odors may help scientists to find ways to interfere with these processes to protect food crops and reduce the spread of tropical diseases.

and suppress responses to large signals (*Bhandawat et al., 2007*; *de Bruyne et al., 2001*). Glomerular mechanisms implement a type of divisive gain control that maintains PN sensitivity within the range of changing ORN responses (*Bhandawat et al., 2007*; *Luo et al., 2010*; *Olsen et al., 2010*; *Olsen and Wilson, 2008*). Finally, transduction currents in response to odor pulses scale inversely with the intensity of the background signal, consistent with the Weber-Fechner Law (*Cafaro, 2016*; *Cao et al., 2016*). However, whether ORN firing follows a similar scaling is unclear (*Cafaro, 2016*; *Martelli et al., 2013*). Thus, although it is known that the input-output curve of ORNs to odor stimuli changes with odor background, how ORN gain (from stimulus to firing rate) scales with background signal intensity has not been characterized.

Olfactory responses in insects can be fast. Transduction can be initiated within milliseconds of odor reaching the antenna (*Szyszka et al., 2014*). The speed of the response is enhanced by ORN spike generation, which emphasizes changes in transduction currents (*Nagel and Wilson, 2011*), and by PNs (*Kim et al., 2015*), which maintain fast information transmission from ORNs to PNs (*Jeanne and Wilson, 2015*; *Nagel et al., 2015*; *Raccuglia et al., 2016*). In contrast, adaptation to high intensity stimuli slows down transduction (*Cao et al., 2016*; *Kaissling et al., 1987*; *Nagel and Wilson, 2011*), a property that might make it difficult to reliably encode the timing of odor encounters.

We investigated in vivo how *Drosophila* ORNs encode encounters with naturalistic odor plumes. To address this question, we first developed an odorant delivery system that reproducibly delivered odorants with naturalistic or Gaussian statistics with controlled means and variances. We simultaneously recorded the odorant stimulus using a fast photo-ionization detector (PID), and recorded extracellularly from identified ORNs.

We found that ORNs encoded broadly distributed naturalistic signals by using two mechanism: front-end nonlinearities that are inherent in receptor binding to ligand, as well as two adaptation mechanisms that are sensitive to the mean and variance of the stimulus. These adaptive mechanisms caused ORNs to rapidly desensitize following encounters with odorant whiffs, dynamically adjusting gain while responding to intermittent odorant stimuli. ORNs adapted to changes in the mean stimulus at the level of transduction by scaling gain inversely with the stimulus intensity, consistent with the Weber-Fechner Law. Variance-dependent gain control took place at both signal transduction and spiking machinery.

While the transduction response time slowed down with increasing stimulus intensity, the spiking machinery sped up to compensate. These complementary kinetic changes caused the firing rate response time to remain invariant with stimulus intensity. This reveals a mechanism that could allow ORNs to preserve information about the precise timing of odor encounters over a wide range of rapidly changing signal intensities. A minimal two-state model of the activity of the Or-Orco complex (olfactory receptor and co-receptor Orco) with an adaptation mechanism that feeds back onto the free energy difference between active and inactive conformations reproduced Weber-Fechner scaling, slowdown of signal transduction kinetics, and responses to intermittent and Gaussian stimuli.

## Results

### ORN responses to naturalistic odorant stimuli show deviations from linearity that arise from adaptation and front end nonlinearities

Odorant signals used to study ORN adaptation typically consist of long pulses or constant backgrounds of various intensities (*Cafaro, 2016*; *Cao et al., 2016*; *Martelli et al., 2013*; *Nagel and Wilson, 2011*). However, airborne stimuli encountered by flying insects can be intermittent with both the intensities of encounters and durations between encounters broadly distributed as power laws (*Celani et al., 2014*). Since ORN transduction can be adapted by odorant pulses as brief as 35 ms on timescales as fast as 500 ms (*Cao et al., 2016*), we asked to what extent the gain of ORNs could change dynamically during responses to naturalistic stimuli, amplifying responses to isolated whiffs of odorant, and suppressing responses to whiffs following dense clumps of whiffs.

We measured the responses of ab3A and ab2A ORNs to naturalistic stimuli of ethyl acetate and 2-butanone. These odorants elicit spikes in these neurons (*Hallem and Carlson, 2006*), and are easy to control and measure (*Martelli et al., 2013*) (*Figure 1—figure supplement 1*). We used in vivo extracellular recording to record both the local field potential (LFP) and spikes from a single sensillum, with simultaneous measurement of the stimulus (*Figure 1—figure supplement 2*). Previous results have shown that: LFP responses are unaffected by the addition of TTX, which eliminates neural spiking, suggesting that LFP signals were generated upstream of the spiking machinery; and that LFP signals are unaffected when the neuron's partner cell in the sensillum is genetically ablated, when that partner does not sense the odorant, suggesting that LFP signals are generated by the neuron of interest (*Nagel and Wilson, 2011*). Though the LFP could reflect activity of nearby sensilla, it serves as an imperfect but useful proxy for transduction activity in ORNs (*Johnston et al., 1995*; *Kaissling, 1986*; *Nagel and Wilson, 2011*; *Su et al., 2012*).

The naturalistic stimulus we used was intermittent and consisted of brief odor whiffs of varied amplitude (*Figure 1a–b*). Durations of whiffs and blanks were broadly distributed, with a power law of exponent $-3/2$ to match natural intermittent statistics of odor plumes (*Celani et al., 2014*) (*Figure 1—figure supplement 3*). ab2A and ab3A ORNs responded to whiffs with transient decreases in the local field potential (LFP) and corresponding increases in the firing rate (*Figure 1a–b*).

Even though individual whiff intensities were broadly distributed (First line in *Figure 1a*) ORN responses to these whiffs were more even, so that responses to faint whiffs were amplified more than those to intense whiffs. To quantify these differences, we defined the gain of the neuron to be the change in the response for a unit change in the stimulus ($gain := \Delta R/\Delta S$). Since ORNs do not respond instantaneously to odorant stimuli (*Cao et al., 2016*; *de Bruyne et al., 2001*; *Martelli et al., 2013*; *Nagel and Wilson, 2011*) we fit linear filters to best predict the LFP and firing rate from the stimulus. We used these filters to make linear predictions of the responses from the stimuli (*Figure 1—figure supplement 4*). Changes in gain were therefore defined as deviations from the linear prediction of response from the stimulus, similar to (*Baccus and Meister, 2002*; *Kim and*

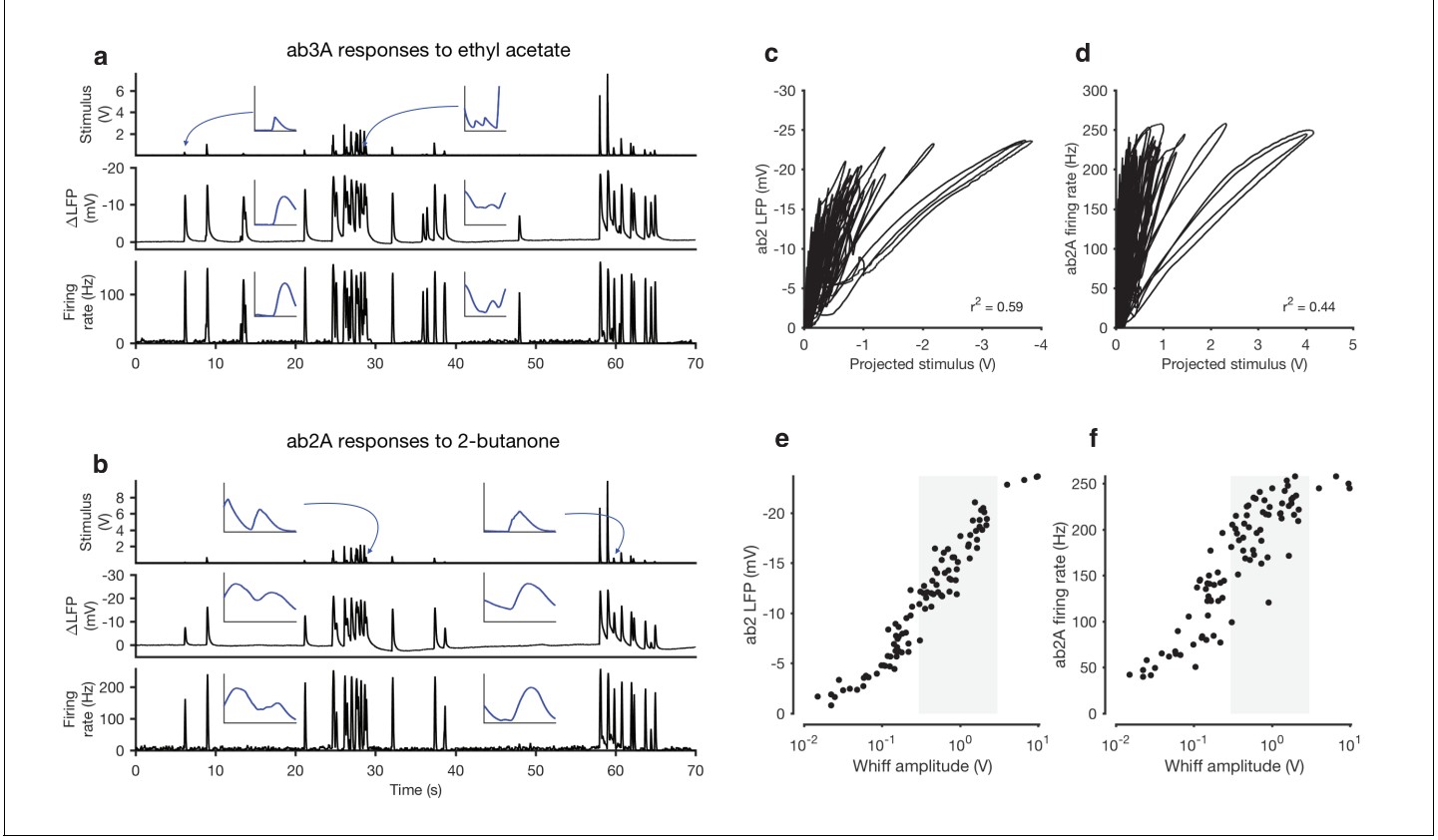

**Figure 1.** Adaptation and saturation modulate ORN responses to broadly distributed naturalistic stimuli. (a) Ethyl acetate odorant (top) elicits LFP (middle) and firing rate (bottom) responses from a ab3A ORN. (b) 2-butanone odorant (top) elicits LFP (middle) and firing rate (bottom) responses from a ab2A ORN. Insets in (a–b) show pairs of whiffs and the LFP and firing rate responses they elicit on an expanded timescale. All pairs of insets are shown at the same scale, for 400 ms around a whiff. (c) ab2 LFP responses *vs.* projected stimulus. (d) ab2A firing rate *vs.* projected stimulus. (c) and (d) show that ORN responses differ significantly from linearity. (e) ab2 LFP responses *vs.* whiff amplitude. (f) ab2A firing rate *vs.* whiff amplitude. n = 15 trials from 2 ORNs. 101 whiffs shown in (e–f).

The following figure supplements are available for figure 1:

**Figure supplement 1.** Diagram of odor delivery device and calibration of Photo-Ionization Detector (PID).

**Figure supplement 2.** Example of simultaneously acquired primary data (ab3A responses to ethyl acetate stimulus).

**Figure supplement 3.** Statistics of the ethyl acetate stimulus with naturalistic temporal structure.

**Figure supplement 4.** Deviations from linearity persist even when filters extracted from Gaussian stimuli are used to project naturalistic stimulus.

*Rieke, 2001*). We visualized these gain changes by plotting the LFP responses against linear prediction of the LFP (*Figure 1c*) and the firing rate against the linear prediction of the firing rate (*Figure 1d*). Each excursion in these plots corresponds to the ORN's response to a single whiff. Excursions occurred with different slopes, suggesting that ORN gain changed frequently in time. Deviations from linearity persisted even when filters computed from Gaussian inputs were used to project the stimulus, suggesting that the existence of these deviations do not depend on the exact shape of the filter, but rather reflect a property of the ORN response not captured by the linear model (*Figure 1—figure supplement 4*).

Variations in the gain ($\Delta R/\Delta S$) clearly do not arise solely from a static output nonlinearity, such as one associated with a linear-nonlinear transformation, since plotting neuron response against the projected stimulus (*Figure 1c–d*) does not yield a single transformative function. (*Dayan and*

*Abbott, 2001*). We reasoned that changes in the gain could arise from *input* nonlinearities due to odor-receptor binding and channel opening. To visualize the nonlinearity between the stimulus and response, we plotted LFP and firing rate responses to each whiff in the naturalistic stimulus as a function of the amplitude of that whiff (*Figure 1e–f*). A clear sigmoidal dependency is visible in the plot of LFP responses against whiff intensity, consistent with a front-end nonlinearity arising from receptor-odorant binding. However, in both the LFP and firing rates, responses to whiffs with similar intensities varied significantly, deviating from a single sigmoidal dose-response curve (*Figure 1e–f*).

What causes these deviations from the dose-response curve? One possibility is that these deviations are due to random variability in the responses of the neuron. Another possibility is that these deviations are due to adaptation of the neuron to the stimulus history preceding each whiff, which may vary with every whiff. To distinguish between these possibilities, we collected whiffs that had similar amplitudes, and examined the LFP and firing rate responses they evoked (*Figure 2a–b*). The amplitude of LFP and firing rate responses elicited by these whiffs varied inversely with the amplitude of the preceding stimulus: whiffs that occurred in isolation (purple) elicited the largest

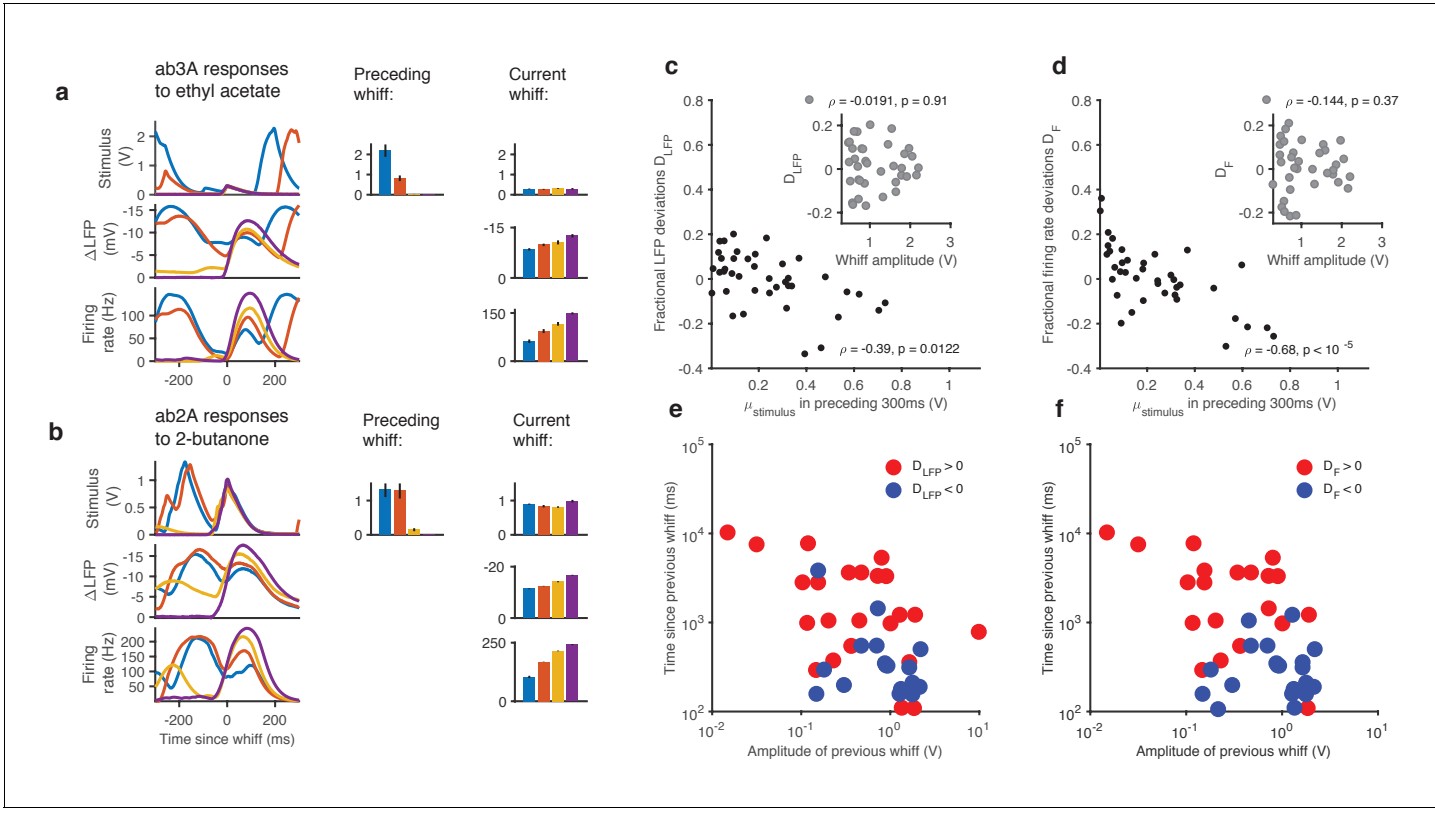

**Figure 2.** Adaptation and saturation modulate ORN responses to broadly distributed naturalistic stimuli. (**a**). Ethyl acetate whiffs of similar size (top) elicit ab3 LFP responses (middle) and ab3A firing rate responses (bottom) with different amplitudes. (**b**) 2-butanone whiffs of similar size (top) elicit ab2 LFP responses (middle) and ab2A firing rate responses (bottom) with different amplitudes. Bar graphs in (**a**) and (**b**) show that ordering in LFP and firing rate response does not correlate with whiff amplitude, but correlates with the intensity of the preceding whiff. Colors on bar graph correspond to colors in time series on the left. Deviations in LFP (**c**) and firing rate responses (**d**) from the median response vs. mean stimulus in the preceding 300 ms. Deviations in LFP (**c**, inset) and firing rate responses (**d**, inset) from the median response vs. whiff amplitude. (**e**) Deviations from the median of LFP responses (positive deviations: red, negative deviations: blue) as a function of the amplitude of the previous whiff and the time since previous whiff. Positive and negative deviations are significantly different ($p = 0.01$, 2-dimensional K-S test). (**f**) Deviations from the median of firing rate responses (positive deviations: red, negative deviations: blue) as a function of the amplitude of the previous whiff and the time since previous whiff. Positive and negative deviations are significantly different, ($p = 0.001$, 2-dimensional K-S test on firing rate deviations).

The following figure supplement is available for figure 2:

**Figure supplement 1.** An NL model (*static* input nonlinearity followed by a linear filter) cannot reproduce context-dependence of LFP responses to similar-sized whiffs.

responses, while whiffs that followed earlier, large whiffs (blue, red) elicited the smallest responses, suggesting that ORN responses can be modulated by stimulus history.

To quantify this context-dependent modulation, we estimated deviations of the LFP and firing rate response to each whiff from the median response (see Materials and methods). Deviations in LFP response to each whiff decreased with mean stimulus in the preceding 300 ms (*Figure 2c*, $\rho = -0.39, \ p = 0.01$, Spearman test), and were uncorrelated with the amplitude of the whiff that elicited them (*Figure 2c*, inset, $p = 0.9$, Spearman test). Similarly, deviations in the firing rate responses to each whiff decreased with mean stimulus in the preceding 300 ms (*Figure 2d*. $\rho = -0.68, \ p<10^{-5}$, Spearman test), and were uncorrelated with the amplitude of the whiff that elicited them (*Figure 2d*, inset, $p = 0.37$, Spearman test).

To generalize beyond a particular timescale of the stimulus history, we parametrized the stimulus history of each whiff by the amplitude and time since the preceding whiff, and grouped estimated deviations from the median response into positive or negative (*Figure 2e–f*). When response deviations were negative (smaller than median responses, blue dots), the amplitude of the preceding whiffs tended to be larger, and the time since the previous whiff tended to be shorter. When response deviations were positive (red dots), the amplitude of preceding whiffs tended to be smaller ($<S_{previous}>$ is 0.52 V when deviations are positive *vs.* 1.09 when deviations are negative for firing rate responses and 1.04 V *vs.* 1.06 V for LFP responses), and the time since the previous whiff tended to be longer ($<t_{previous}>$ is 3025 ms *vs.* 361 ms for firing rate responses, and 2556 ms *vs.* 566 ms for LFP responses).

What causes this context-dependent suppression of responses following preceding whiffs? One possibility is a bi-lobed filter, with one positive and one negative lobe, followed by a rectifying non-linearity. Such a filter is partly differentiating, and has been measured in linear models of the firing rate (*Kim et al., 2011*; *Martelli et al., 2013*; *Nagel and Wilson, 2011*) and would lead to attenuated responses to the second of two closely spaced whiffs due to linear superposition. Such a mechanism may partly account for context dependent variation in firing rates. However, stimulus-to-LFP filters, computed for this stimulus and others, are mono-lobed (*Figure 2—figure supplement 1*), and appear purely integrating (*Nagel and Wilson, 2011*), ruling out contributions to dynamic modulation of LFP responses by this mechanism. A model with a static front-end nonlinearity and a mono-lobed filter fit to the LFP also cannot reproduce context-dependent adaptation observed in the LFP (*Figure 2—figure supplement 1*), suggesting that this context-dependent variation in response arises at least in part from ORNs dynamically varying gain in response to naturalistic stimuli.

Since the mean and variance of naturalistic stimuli are correlated over many timescales, (*Figure 1—figure supplement 3*), it is unclear whether adaptation in this context is sensitive to the mean or the variance (or to some other statistic) of preceding whiffs. To determine how changing one moment of the stimulus distribution changed ORN gain, and to disambiguate the effect of receptor saturation from adaptation, we proceeded to other experiments using Gaussian stimuli with changing means (*Figure 3*) and variances (*Figure 4*).

## ORNs adapt to stimulus background by decreasing gain according to the Weber-Fechner Law

A common strategy used by sensory systems to encode signals over a broad range of background intensities is to scale the response according to the Weber-Fechner law (*Fechner, 1860*; *Stevens, 1957*; *Weber, 1834*), that is, to control gain inversely with stimulus mean. In ORNs, transduction currents elicited by odorant pulses are reduced by preceding pulses (*Nagel and Wilson, 2011*) and scale inversely with background intensity, consistent with the Weber-Fechner law (*Cafaro, 2016*; *Cao et al., 2016*). It remains unclear whether the ORNs' ultimate output — the firing rate — follows the same Weber-Fechner scaling (*Cafaro, 2016*; *Martelli et al., 2013*). We therefore stimulated ab3A ORNs with a set of fluctuating ethyl acetate stimuli with increasing means (*Figure 3a*) but roughly constant variances (*Figure 3b*, *Figure 2—figure supplement 1a*). ORNs responded to the stimulus with the smallest mean by modulating firing rates between 0–60 Hz (*Figure 3c*). This response range progressively decreased with increasing mean stimulus intensity (*Figure 3d*), though the mean response remained at ~30 Hz. To estimate ORN input-output curves, we plotted ORN responses against the stimulus projected through the normalized best-fit linear filter for each stimulus, estimated by least-squares fitting (*Baccus and Meister, 2002*; *Chichilnisky, 2001*; *de Boer and*

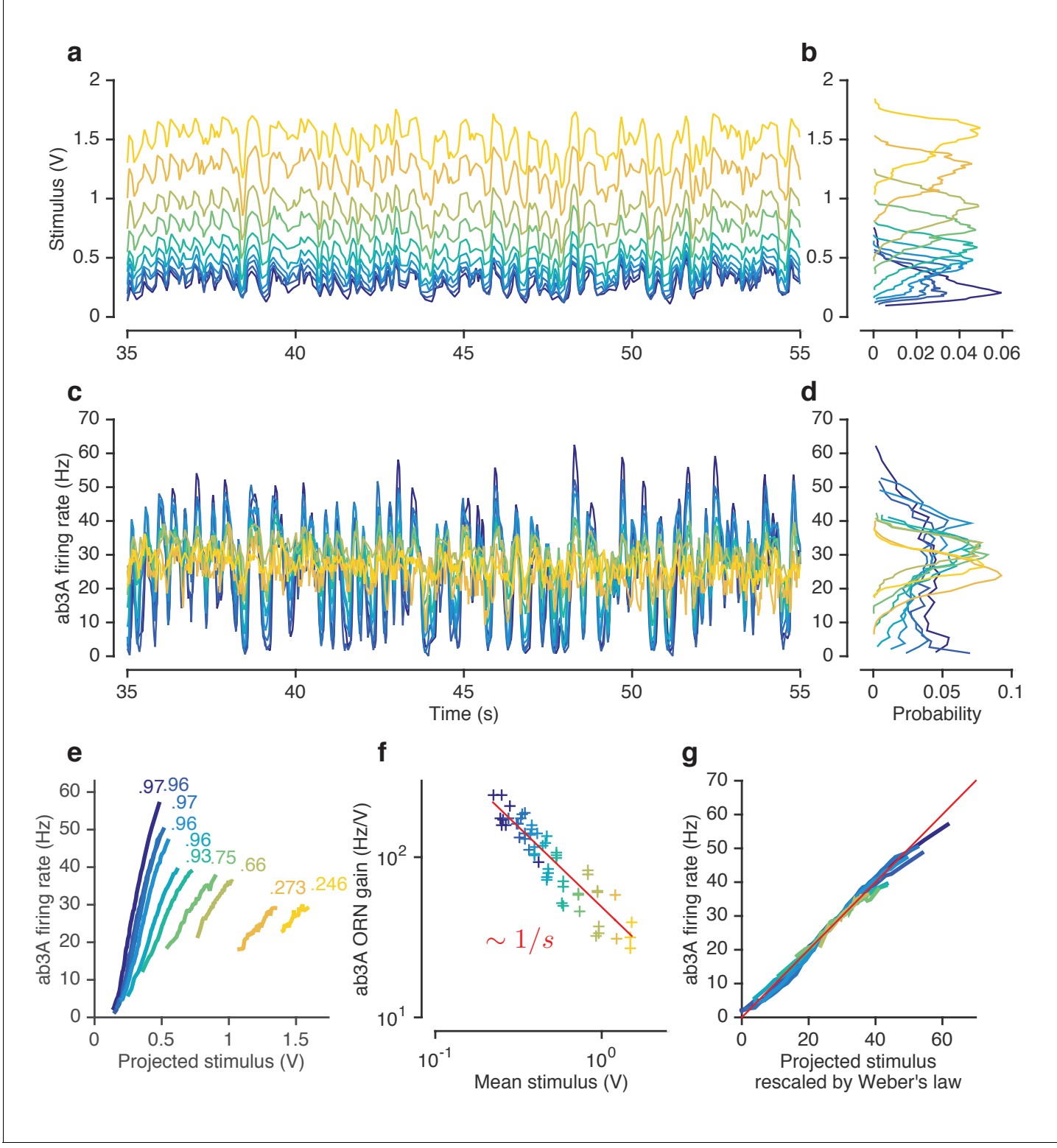

**Figure 3.** ORNs decrease gain with stimulus mean, consistent with the Weber-Fechner Law. (a) Ethyl acetate stimuli with different mean intensities but similar variances. Stimulus intensity measured using a Photo-Ionization Detector (PID), units in Volts (V). Colors indicate mean stimulus intensity. (b) Corresponding stimulus distributions. (c) ab3A firing rate responses to these stimuli. (d) Corresponding response distributions. (e) ORN responses vs. stimulus projected through linear filters. Colored numbers indicate $r^2$ between linear projections and ORN response. (f) ORN gain vs. mean stimulus for each trial. Red line is the Weber-Fechner prediction ($\Delta R/\Delta S \sim 1/S$) (g) After rescaling the projected stimulus by the gain predicted by the red curve in (f),

*Figure 3 continued on next page*

*Figure 3 continued*

and correcting for an offset, ORN responses collapse onto one line. n = 55 trials from 7 ORNs in 3 flies. All plots except (**f**) show means across all trials. (**f**) shows individual trials.

The following figure supplements are available for figure 3:

**Figure supplement 1.** Weber-Fechner Law broadly observed across odor-receptor combinations.

**Figure supplement 2.** Ability of NL models to reproduce observed change in input-output curves.

**Figure supplement 3.** Projected stimulus rescaled by Weber-Fechner relation correlate with firing rates.

*Kuyper, 1968*; *Rieke et al., 1997*) (*Figure 3e*) Input-output curves grew shallower with increasing mean stimulus. We defined the ORN gain to be the slope of the input-output curve at that mean stimulus, similar to (*Baccus and Meister, 2002*).

ORN gain in each trial varied with the mean stimulus in that trial as an approximate power law with exponent –1 (*Figure 3f*). ORN gains could also be estimated by the ratio of standard deviation of the response to the standard deviation of the stimulus. This measure yielded similar values of ORN gain, and also decreased as a power law with exponent −1 (*Figure 3—figure supplement 1b*). This exponent is consistent with the Weber-Fechner Law, which postulates that the just noticeable difference between two stimuli is inversely proportional to the stimulus magnitude (*Stevens, 1957*). Rescaling the projected stimulus by the gain predicted by Weber's Law collapsed all input-output curves onto a single curve (*Figure 3g*).

Can front-end or back-end nonlinearities reproduce the observed change of input (stimulus)-output (firing rate) curves (*Figure 3e*)? Clearly, no single *output* nonlinearity can fit the data shown in (*Figure 3e*), since a single function cannot fit all the input-output curves. Since a front-end nonlinearity is present (*Figure 1*), we asked whether a static nonlinear-linear (NL) model could reproduce this data, with the input nonlinearity parameterized by a Hill function $S/(S + K)$ where $S$ represents the input, and $K$ the half maximum value. (*Figure 3—figure supplement 2a–c*). NL model responses increased with mean stimulus (*Figure 3—figure supplement 2c*), unlike in the data (*Figure 3d–e*). However, if the half maximum value of the Hill function was allowed to vary with the mean stimulus, the model could qualitatively reproduce the data, suggesting adaptation at the front-end nonlinearity (*Figure 3—figure supplement 2d–f*).

To determine if similar gain-control relative to mean signal intensity was broadly observed, we tested additional ORNs from the two major olfactory organs of the fly, the antenna and the maxillary palp (ab2A, pb1A), and used ecologically relevant odorants from three different functional groups (ketones: 2-butanone, alcohols: 1-pentanol, esters: isoamyl acetate) in various combinations. In all five cases, the neurons decreased gain with increasing odorant concentration, and obeyed a roughly inverse scaling (*Figure 3—figure supplement 1c–f*). Rescaling the projected stimulus by the Weber-Fechner relation collapsed all ORN responses onto a single curve, similar to *Figure 3g* (*Figure 3—figure supplement 3*). Thus in vivo, for various neurons and odorants, ORN firing rate followed the Weber-Fechner Law.

## Fast variance-dependent gain control in ORNs

In other sensory modalities, such as vision, some peripheral neurons adapt not only to the mean but also to the variance of the signal (*Baccus and Meister, 2002*; *Rieke, 2001*). We therefore asked whether ORNs adjust their gain in response to changes in the variance of the signal. We stimulated ab3A ORNs with fluctuating ethyl acetate stimuli in which the variance of the signal changed every 5 s (*Figure 4a*), switching back and forth between high to low values, around a nearly constant mean (*Figure 4b*; *Figure 4—figure supplement 1a*), a protocol used to study gain control in visual neurons (*Baccus and Meister, 2002*; *Fairhall et al., 2006*; *Shapley and Victor, 1978*; *Rieke, 2001*; *Smirnakis et al., 1997*).

As expected, ORNs responded to input fluctuations by modulating their firing rate. Interestingly, ORN firing rate variance did not vary as much as the stimulus variance between epochs of high and low stimulus variances, suggesting that ORNs actively changed their gain to compensate for such

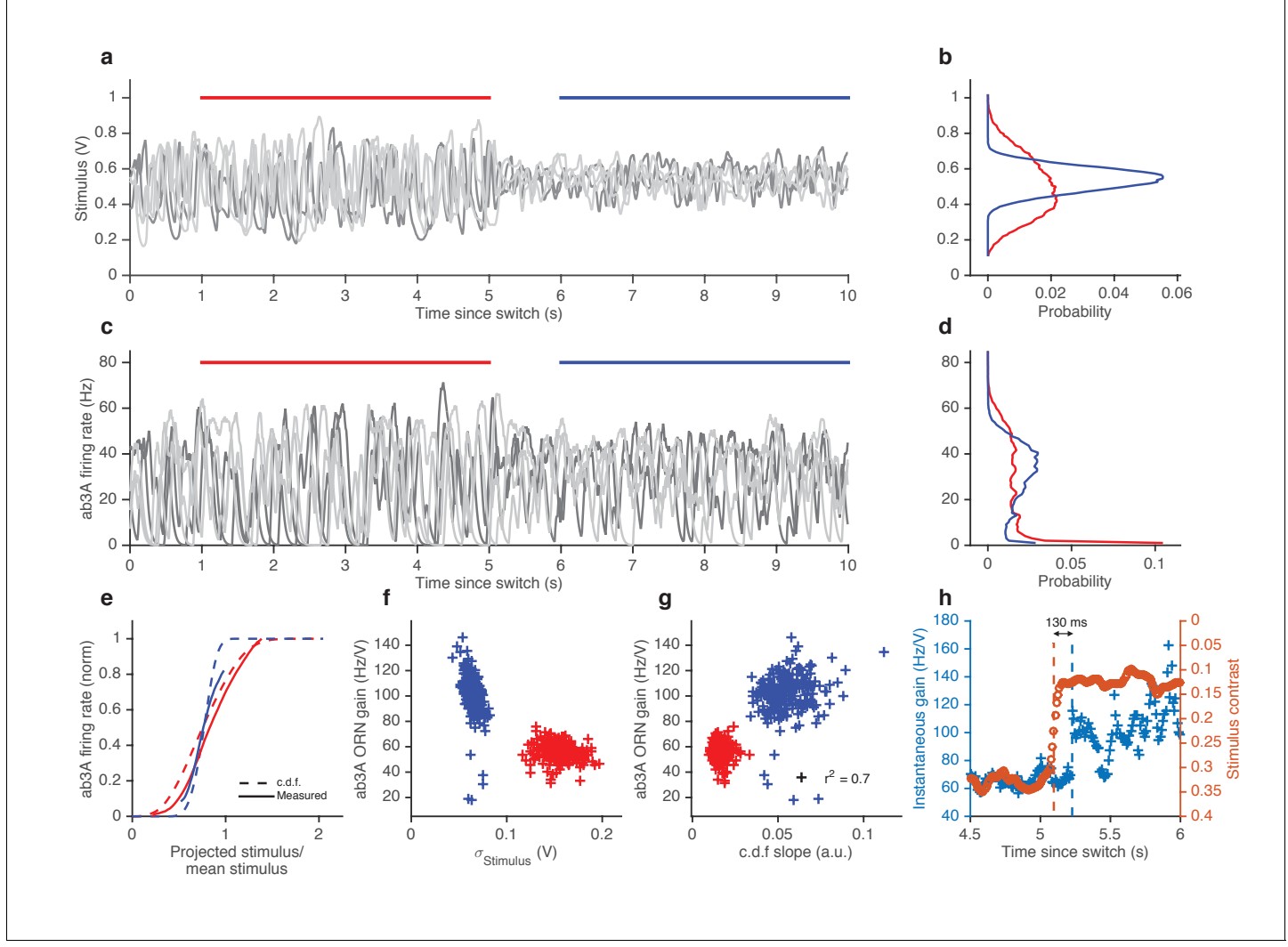

**Figure 4.** ORNs decrease gain with stimulus variance. (a). Stimulus intensity of a fluctuating ethyl acetate stimulus with nearly constant mean but a variance that switches between high and low every 5 s. Five independent trials (out of 248) are plotted. (b) Distributions of stimulus intensity for the epochs of low (blue) and high (red) variance. (c) ab3A firing rate responses corresponding to the trials shown in (a) following the switch from low to high variance, which takes place at t = 0 s and from high to low, which takes place at t = 5 s. (d) Probability distributions of the response. (e) Solid lines are ORN input-output curves computed from a single filter from both low (blue) and high (red) variance epochs. Dashed lines are the cumulative distribution functions (c.d.fs) of the projected stimulus. (f) ORN gain as a function of the standard deviation of the stimulus, measured per trial for each epoch. (g) Measured gain plotted against the slope of the cumulative distribution function for each trial. (h) Instantaneous gain (blue) and stimulus contrast (orange) as a function of time since switch. Dashed lines indicate crossover times of stimulus contrast and instantaneous gain. The delay is ~130 ms. n = 248 trials from 5 ab3A ORNs in 2 flies.

The following figure supplement is available for figure 4:

**Figure supplement 1.** Variance gain control in Gaussian stimuli.

input differences (*Figure 4c–d*). ORN input-output curves during high variance epochs (red) were shallower than during low variance (blue) epochs (*Figure 4e*). Trial-wise ORN gain decreased with the variance of the stimulus (*Figure 4f*). ORN gains estimated by dividing the standard deviation of the response by the standard deviation of the stimulus showed a similar decrease in ORN gain with stimulus variance (*Figure 4—figure supplement 1b*).

A simple coding strategy maximizes a neuron's information capacity by matching its input-output curve to the cumulative distribution function (c.d.f) of the stimulus (*Laughlin, 1981*). Like the c.d.f.s

(dashed lines), the input-output curves (solid) are steeper during the low variance epoch. On a trial-by-trial basis, ORN gain was correlated with the c.d.f slope ($r^2 = 0.7$) (*Figure 4g*). However, as the input variance changed by a factor of 2.5, the gain in the neuron only changed by a factor of 1.7, not as much as would be required for optimal information encoding. In these experiments, the gain changed within ~130 ms following the change in stimulus variance (*Figure 4h*).

## Mean and variance gain control occur at different stages of odor encoding, and are mechanistically distinct

ORN responses arise through two sequential steps: odor transduction followed by spike generation (*Nagel and Wilson, 2011*). Does each step possess separate gain control mechanisms, or is gain control achieved solely at one step? Previous studies place the mechanism of adaptation to mean stimulus at the level of signal transduction (*Cafaro, 2016*; *Cao et al., 2016*; *Nagel and Wilson, 2011*). How the spiking machinery might influence gain control, and where adaptation to signal variance takes place, remain unknown.

We reanalyzed the responses of ab3A to ethyl acetate signals (*Figure 2 and 3*) and measured 'transduction gain' (stimulus to LFP) and 'firing gain' (LFP to firing rate). Changing the stimulus mean alone changed gain in LFP (*Figure 5a–b*, *Figure 5—figure supplement 1*). However, gain at the spiking machinery was largely invariant to the ten-fold change in the mean stimulus (p=0.41, Spearman rank correlation), with a 1 mV change in LFP leading to a ~ 10 Hz change in the firing rate, consistent with earlier studies (*Nagel and Wilson, 2011*) (*Figure 5c–d*). Transduction gain, like ORN gain, scaled with the Weber-Fechner Law, for a variety of odor-receptor combinations (*Figure 5—figure supplement 1a-d*) consistent with previous studies (*Cafaro, 2016*; *Cao et al., 2016*). In contrast, adaptation to the stimulus variance changed gain both at transduction and at spiking (*Figure 5e–h*). Both gains changed by a factor of ~1.3 from the high to the low variance epoch (p<*0.001*, Wilcoxon signed rank test), contributing roughly equally to variance gain control (*Figure 5—figure supplement 1f*).

## Modularity of gain control at transduction and spiking

Stimulating ORNs with odorants evokes responses from both transduction and spiking machinery, making it hard to determine how independent gain control at the two modules are. To further pinpoint the contributions of signal transduction and firing machineries to gain control, we expressed Chrimson channels (*Klapoetke et al., 2014*) in ab3A ORNs and activated them using red light, either in isolation or in combination with odorants.

First, we used a fluctuating ethyl acetate stimulus to probe transduction and ORN gain while increasing the neuron's firing rate using increasing backgrounds of red light (*Figure 6a–b*). While increasing light levels elicited increasing firing rates (*Figure 6b* inset), ORN and transduction gain did not vary with the intensity of supplemental light (*Figure 6a–b*). This suggests that constitutive spiking activity does not feed back onto LFP adaptation, or overall ORN gain.

Second, we used a fluctuating light stimulus to probe the spiking gain while stimulating the ORN and its receptors with increasing backgrounds of ethyl acetate odorant (*Figure 6c–d*). While ethyl acetate backgrounds of increasing intensity increased ORN firing rate (*Figure 6d* inset), they failed to change gain in the spiking machinery. Increasing odor backgrounds moved input-output curves along the y-axis (*Figure 6c*), consistent with increasing firing due to background odor, but failed to change the slope of these curves, suggesting that ORN gain to the fluctuating light probe was not changed. This suggests that adaptation at transduction does not affect gain of the spiking machinery, consistent with our result that increasing odor backgrounds decreased gain at transduction, but not spiking (*Figure 5a–d*). Thus, Weber-Fechner scaling in ORN gain control to stimulus mean is insulated from activity of the spiking machinery.

Variance gain control exists in a wide range of neurons (*Baccus and Meister, 2002*; *Díaz-Quesada and Maravall, 2008*; *Nagel and Doupe, 2006*; *Rieke, 2001*; *Wark et al., 2009*; *Zaghloul et al., 2005*) and in models of spiking neurons (*Gaudry and Reinagel, 2007*; *Hong et al., 2007*; *Yu and Lee, 2003*; *Yu et al., 2005*), suggesting that variance gain control could be an intrinsic property of spiking neurons. To determine if the spiking machinery alone could give rise to variance gain control, we stimulated ab3A ORNs that express Chrimson with fluctuating light stimuli of different variances at fixed mean. ORN input-output curves were steeper when the variance of the light

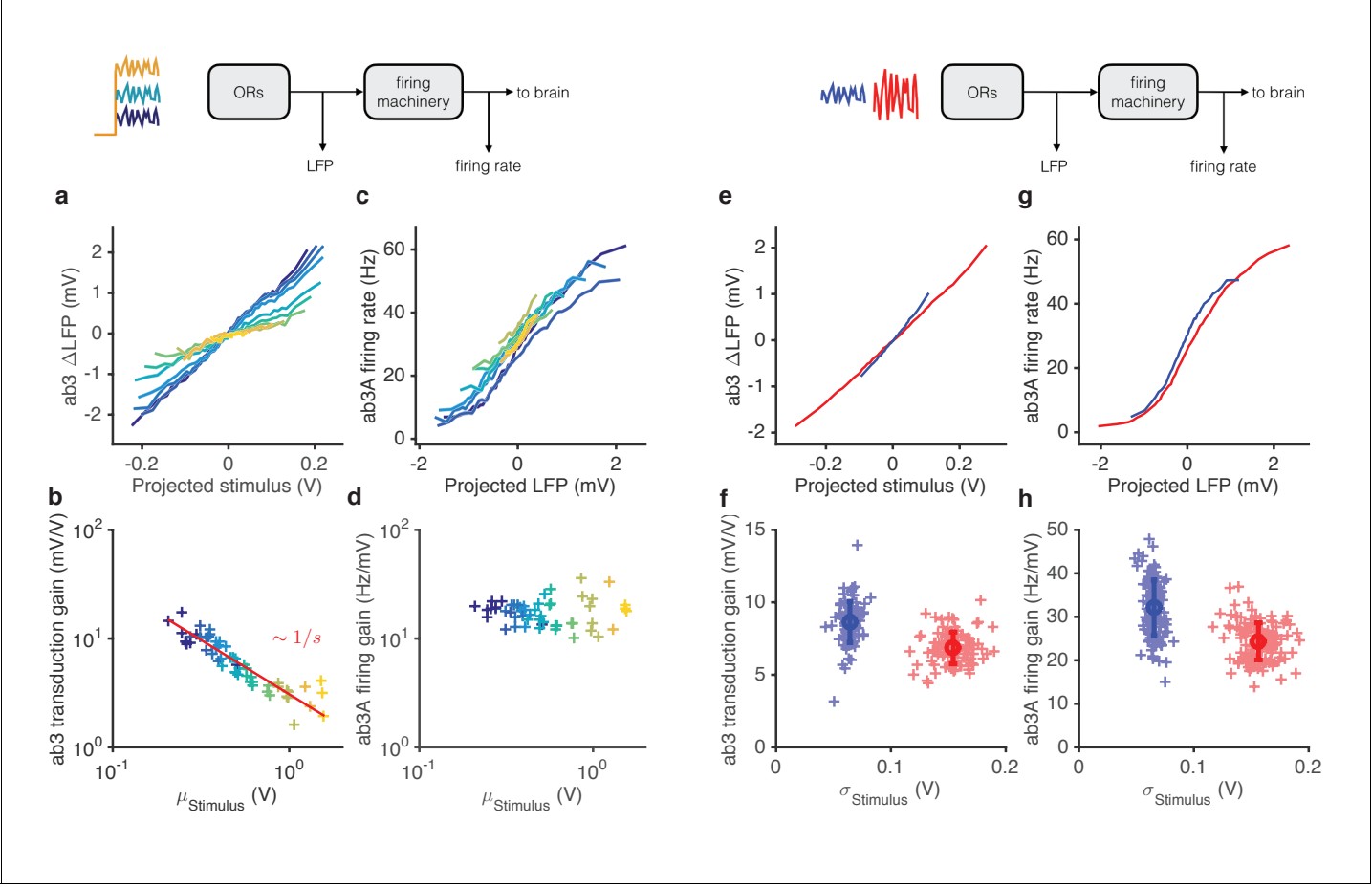

**Figure 5.** Mean gain control occurs primarily at transduction, and variance gain control occurs both at transduction and at the firing machinery. (**a**) Transduction input-output curves from stimulus to LFP. Colors indicate increasing mean stimulus. Filters and projections are computed trial by trial. (**b**) Transduction gain, measured from the slopes of these input-output curves, decreases with the mean stimulus. The red line is a power law with exponent −1, (Weber's Law). (**c**) Input-output curves for the firing machine module. (**d**) Firing gain does not change significantly with mean stimulus. (**e**) Transduction input-output curves for low (blue) and high (red) variance stimuli. (**f**) Transduction gains in the low variance epoch are significantly higher than transduction gains in the high variance epoch (p<0.001, Wilcoxon signed rank test) (**g**) Input-output curves of firing machinery during low variance stimuli. (**g**) Firing gain during low variance epochs are significantly higher than firing gains during high variance epochs (p<0.001, Wilcoxon signed rank test). Projections of stimulus are divided by the mean stimulus in each trial to remove the small effect Weber-Fechner gain scaling. Data in this figure is same as in *Figures 3* and *4*. (**a,c,e,g**) Mean across all trials. (**b,d,f,h**) Individual trials.

The following figure supplement is available for figure 5:

**Figure supplement 1.** LFP responses to fluctuating Gaussian ethyl acetate signals with increasing mean.

stimulation was smaller (*Figure 6e–f*), similar to the curves observed with odor stimulation (cf. *Figure 4e*). We observed that gain changed by a factor of ~1.5 when the standard deviation of the light stimulus changed by a factor of ~3, consistent with variance gain control occurring partly in the spiking machinery (*Figure 5e–h*), though Chrimson channels might exhibit their own nonlinear activation properties.

## Despite slowdown in transduction, ORN firing rate preserves timing of odor encounters

When navigating odor plumes, the precise timing of the encounter with the plume carries important information, which may be lost if adaptation changes the lag between signal and response in a concentration-dependent manner. The kinetics of ORN spiking in response to pulses of odorant are

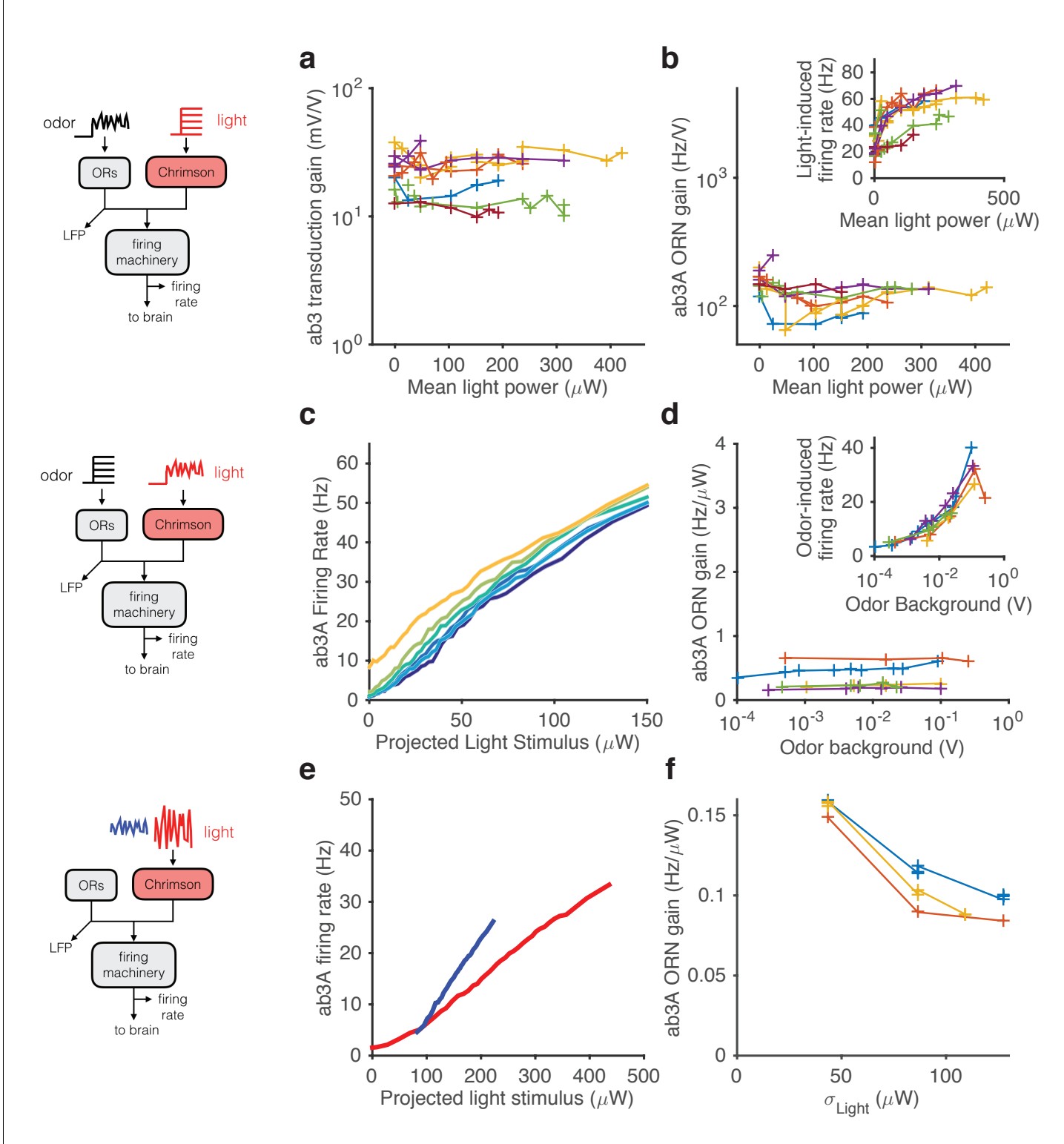

**Figure 6.** Modularity of gain control revealed by optogenetic stimulation. ab3A ORNs in w; 22a-GAL4/+; UAS-Chrimson/+ flies can be activated by ethyl acetate odorant or by red light. (**a–b**) Fluctuating odor foreground and constant light background. (**a**) Transduction gain to fluctuating odor *vs.* background light stimulation intensity. (**b**) Overall ORN gain to fluctuating odor stimulus *vs.* background light stimulation intensity. (b, inset) ORN firing rate *vs.* background light intensity. (**c–d**) Fluctuating light foreground and constant odor background stimulus. (**c**) Input-output curves to fluctuating light stimulus for increasing background odor (lighter colors indicate larger odor background). (**d**) ORN gain is invariant with background odor concentration. (d, inset) Odor-induced firing gain *vs.* background odor concentration. (**e–f**) Fluctuating light stimulus with different variances. (**e**) Input-output curves for

*Figure 6 continued on next page*

*Figure 6 continued*

high (red) and low (blue) variance light stimuli. (**f**) ORN gain as a function of the standard deviation of the light stimulus. (**a–b**) n = 75 trials from 13 ORNs. (**c–d**) n = 64 trials from 5 ORNs. (**e–f**) n = 21 trials from 3 ORNs. Lines link trials from a single ORN.

invariant to the pulse intensity and to the background intensity over a range of odorant concentrations (*Martelli et al., 2013*). Paradoxically, adaptation to background odorants slows transduction current responses to odor pulses (*Cao et al., 2016*; *Kaissling et al., 1987*; *Nagel and Wilson, 2011*). We hypothesized that these seemingly contradictory results might be resolved if the ORN spiking machinery speeds up to compensate for the intensity-dependent slowdown in the LFP.

We characterized responses to odorant stimuli on increasing backgrounds by measuring both ORN spike rates and LFPs. We computed cross correlation functions between the stimulus and the LFP for various stimulus backgrounds. For stimuli on low backgrounds, LFP cross-correlation functions peaked earlier, while for stimuli on larger backgrounds, LFP cross-correlation functions peaked later (*Figure 7a*), consistent with previous results (*Cao et al., 2016*; *Kaissling et al., 1987*; *Nagel and Wilson, 2011*). Surprisingly, cross-correlation functions from the stimulus to the firing rate were similar between stimuli on low and high backgrounds (*Figure 7b*), consistent with

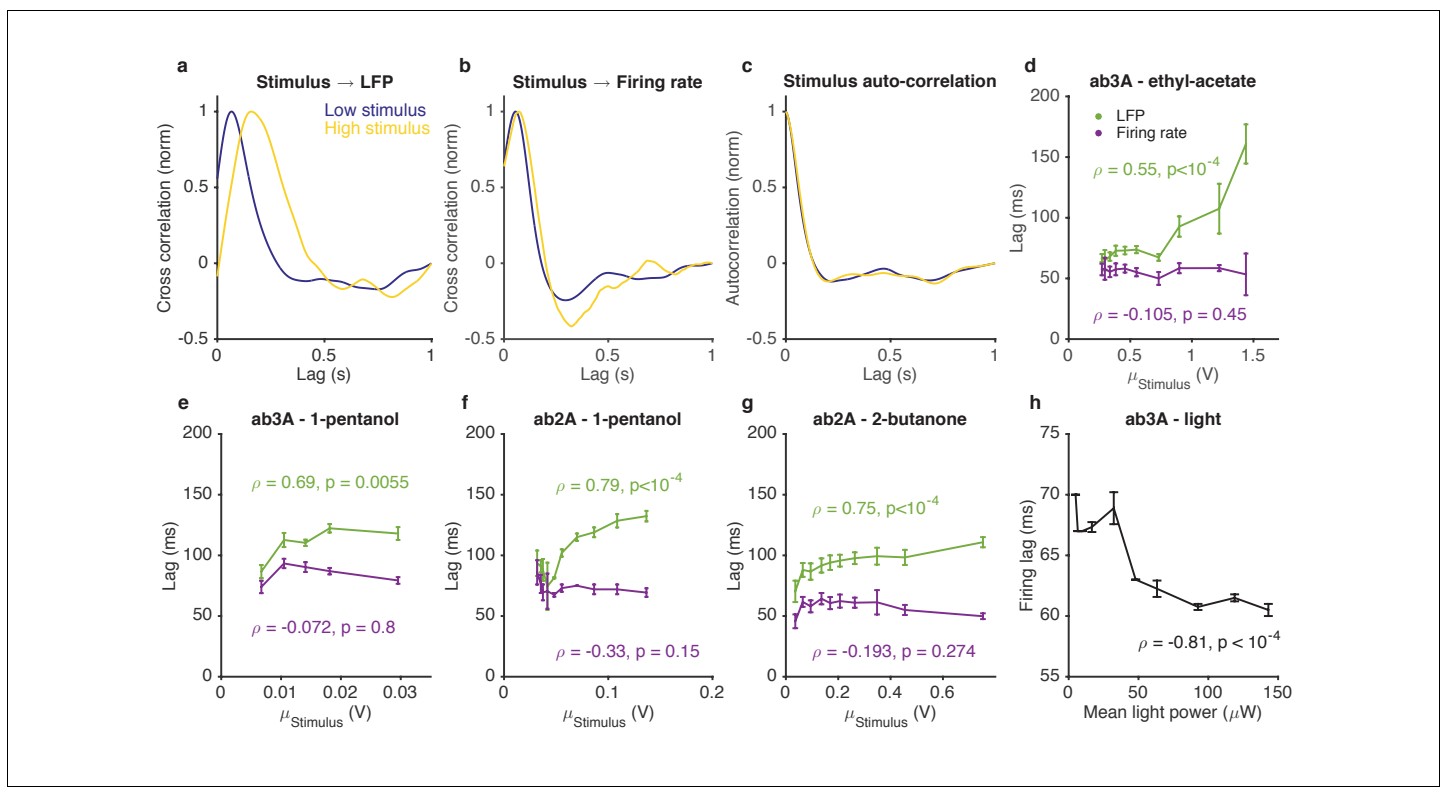

**Figure 7.** Adaptation to the mean slows down LFP, but not firing rate. (**a–d**). Response of ab3A ORNs to Gaussian ethyl acetate stimuli on increasing backgrounds. (**a**) Cross correlation functions between ethyl acetate stimulus and ab3 LFP responses for low (purple) and high (yellow) background stimuli. (**b**) Cross correlation functions between ethyl acetate stimulus and ab3A firing rate responses for low (purple) and high (yellow) background stimuli. (**c**) Stimulus autocorrelation functions for low (purple) and high (yellow) background stimuli. (**d–g**) LFP and firing rate lags with respect to the stimulus vs. the mean stimulus for various odor-receptor combinations. LFP lags increase with mean stimulus, while firing rate lags do not. (**h**) Firing lags of ab3A ORNs expressing Chrimson channels vs. applied light power. In (**c–g**), $\rho$ is the Spearman correlation coefficient, and p is the corresponding p-value.

The following figure supplement is available for figure 7:

**Figure supplement 1.** Variance gain control does not change response kinetics.

(*Martelli et al., 2013*). This selective change in the kinetics of the LFP, but not the firing rate, occurred even though there was no change in the stimulus autocorrelation function from low to high stimulus (*Figure 7c*). We defined the LFP and firing rate lags relative to the stimulus by the location of the peak of the cross-correlation function, and found that while LFP response lags increased with increasing odorant concentration ($p<10^{-2}$, Spearman test), firing rate lags remained relatively invariant with odorant concentration ($p>0.1$, Spearman test) (*Figure 7d–g*).

For firing rate lags to remain invariant with odorant concentration despite a slowdown in transduction, the kinetics of the spiking machinery need to speed up with increasing input to the cell. To test if this is the case, we stimulated ab3A ORNs expressing Chrimson with Gaussian red light stimuli with increasing means, and measured lags between the applied light stimulus and firing rate. Firing lags decreased with increasing light power concentration ($p<10^{-4}$, Spearman test), suggesting that the ORN spiking machinery can speed up with increasing mean input currents (*Figure 7h*), as would accompany increasing odor backgrounds (*Cao et al., 2016*).

If adaptation to the mean slows down transduction, which is compensated for at spiking, does adaptation to the stimulus variance also lead to similar compensatory kinetics? We found that a three-fold change in the stimulus variance, despite leading to changes in LFP and firing gains (*Figures 4–5*), did not significantly change kinetics either at LFP or firing rate (*Figure 7—figure supplement 1*), consistent with our earlier results suggesting that mean and variance gain control have distinct mechanisms.

## An adaptive two-state receptor-complex model reproduces Weber-Fechner scaling, slow down of LFP kinetics, and responses to intermittent and Gaussian stimuli

How do adaptive mechanisms at transduction preserve both the Weber-Fechner Law and lead to response slowdowns? In the following we show that a minimal two-state model of the olfactory receptor-olfactory co-receptor (Or-Orco) complex with an adaptation architecture similar to that of the bacterial chemotaxis system (*Asahina et al., 2009*; *Barkai and Leibler, 1997*; *Emonet and Cluzel, 2008*; *Shimizu et al., 2010*) can reproduce the LFP responses to naturalistic and Gaussian stimuli, as well as Weber-Fechner Law and its accompanying response slow down.

In our model, Or-Orco complexes can be active or inactive (*C* and *C** in *Figure 8a*) and the active complex binds odorant *S* with higher affinity than the inactive complex. We assume that ligand (un) binding is fast compared to (in)activation rates ($w_+$ and $w_-$ in *Figure 8b*). The fraction *a* of active complexes therefore obeys the equation

$$\frac{da}{dt} = (1-a)\,w_+(S,\varepsilon) - a\,w_-(S,\varepsilon) \tag{1}$$

where the rates of activation $w_+(S,\varepsilon)$ and inactivation $w_-(S,\varepsilon)$ are nonlinear functions of the odor concentration *S* and of the free energy difference $\varepsilon$ between the unbound active and inactive states (*Equations 3-4* in Materials and methods). The LFP is modeled as a linear filter acting on the activity (*Figure 8b*, *Equation 5* in Materials and methods). At steady state, *Equation (1)* reduces to $\bar{a}(S,\varepsilon) = 1/(1 + w_-(S,\varepsilon)/w_+(S,\varepsilon))$, where the bar indicates steady state. $\bar{a}(S,\varepsilon)$ is a monotonically increasing function of the odor concentration *S*. Increasing the free energy difference $\varepsilon$ shifts this function towards higher values of *S*, therefore reducing the sensitivity of the system. We model adaptation by assuming that activity of the Or-Orco complex controls the activity of factors that act on the complex to modify the free energy difference $\varepsilon$:

$$\frac{d\varepsilon}{dt} = \beta(a - a_0) \tag{2}$$

where $\beta$ is the rate of adaptation. Importantly, the rate of change of $\varepsilon$ only depends on the activity *a* but not on the free energy difference $\varepsilon$. The architecture of this feedback is similar to that of the bacterial chemotaxis system and ensures that for increasing values of *S*, the changes in $\varepsilon$ compensate for changes in free energy due to ligand binding (*Barkai and Leibler, 1997*). Thus, adaptation eventually returns *a* to the adapted value $a_0$ providing Weber-Fechner scaling (*Shimizu et al., 2010*) (as in *Figure 3*). We assume that the free energy of the complex can only be changed within a finite range, and that the lower bound $\varepsilon_L$ is reached for small values of *S*. Thus, in the absence of ligand,

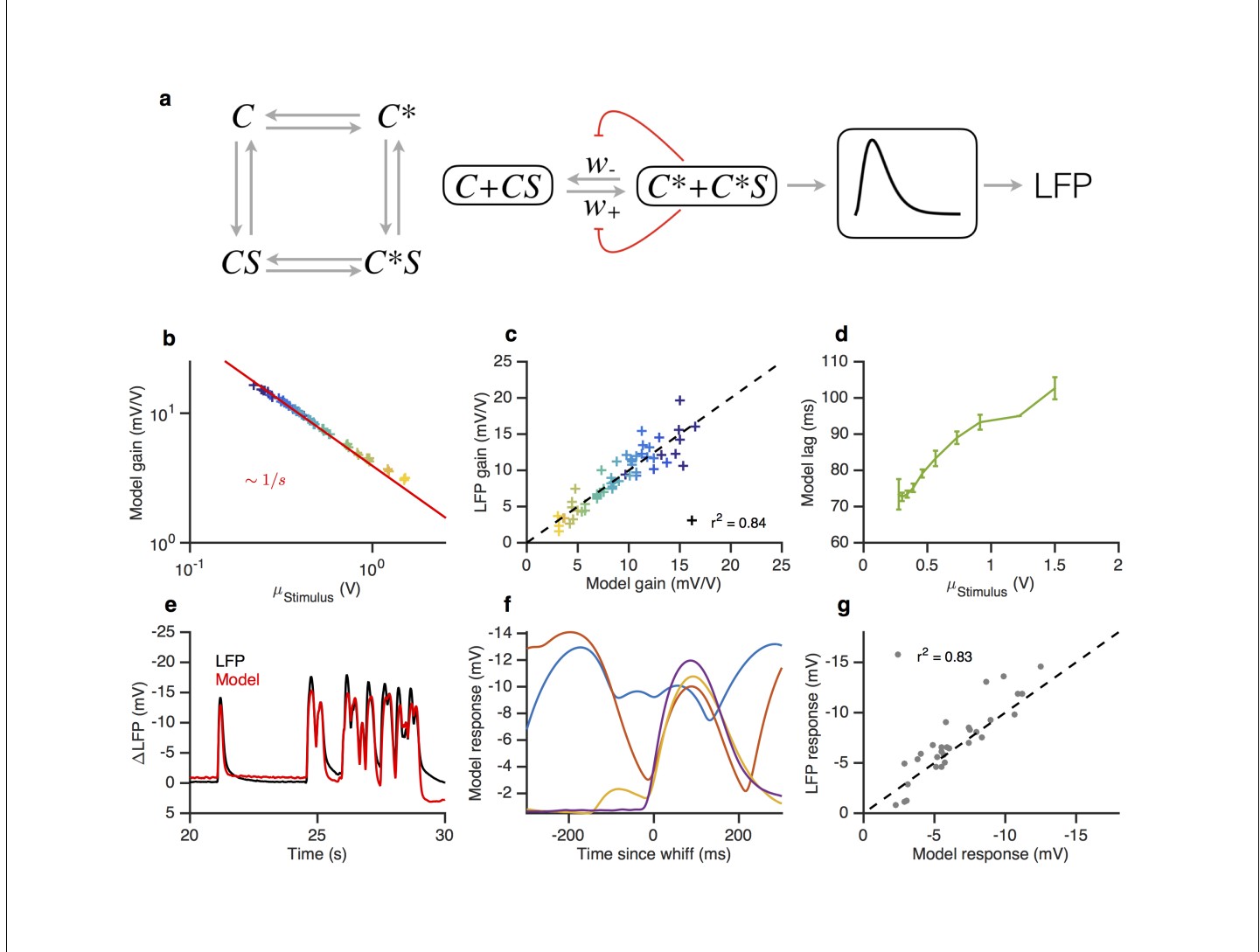

**Figure 8.** A modified two state receptor model reproduces Weber's Law and adaptive slowdown in LFP responses. (**a**). Or-Orco complexes (**C**) can be bound or unbound and active or inactive. (**b**) We assume (un)binding rates are much faster than (in)activation rates. Activity of the complex feeds back onto the free energy difference between active and inactive conformations, which also decreases the activation and inactivation rates of the complex (**Equations 1–4**). A mono-lobed filter converts receptor activity into LFP signals (**Equation 5**). We fit the model to Gaussian (**Figures 3** and **5**) and naturalistic data (**Figures 1–2**). In these fits, $\alpha = 12.5\ s^{-1}$, $\beta = 1.26\ s^{-1}$, $\varepsilon_L = 0.86$, $K_{on} = 0.1\ V$ and $K_{off} = 400\ V$. (**c**) Model gain vs. mean stimulus. Red line is the Weber-Fechner prediction ($\Delta R/\Delta S \sim 1/S$). (**c**) LFP gain vs. model gain. (**e**) Model response lag with respect to stimulus vs. mean stimulus. (**f**) LFP and model responses to naturalistic stimulus. (**g**) The model reproduces LFP responses to similar-sized whiffs that vary inversely with the size of preceding whiffs. (cf. **Figure 2**). (**h**) LFP responses vs. model responses for every whiff in the naturalistic stimulus.

The following figure supplements are available for figure 8:

**Figure supplement 1.** Steady state activity as a function of the stimulus background.

**Figure supplement 2.** Front-end adaptation followed by a LN model reproduces firing rate responses to Gaussian and naturalistic stimuli.

the steady state activity can be smaller than $a_0$. For non-zero values of $S$, the steady state activity first increases with background signal intensity (**Martelli et al., 2013**), before it becomes independent of background intensity once it reaches $a_0$ (**Figure 8—figure supplement 1**), as seen in **Figure 3c**.

An important intrinsic property of this model is that adaptation to increasing background of odorant decreases the rates of activation and inactivation, $w_+$ and $w_-$, of the Or-Orco complex ,

providing a self-consistent explanation for the slowdown of the response kinetics of the LFP upon adaptation. It is interesting to note that this kinetic property emerges because (1) the switching rates are decreasing functions of the free energy difference $\varepsilon$ and (2) the requirement of Weber-Fechner scaling, which causes the adapted value of $\varepsilon$ to scale with the logarithm of the mean signal intensity (see Materials and methods).

The resulting model (*Equations 1–2*, and *Equations 3–5* in Materials and methods) contains six parameters plus another three for converting the signal from activity to LFP. We fit this model to LFP responses to the Gaussian and naturalistic stimuli. The model decreased gain with the mean stimulus background, consistent with Weber-Fechner Law (*Figure 8c*), and predicted the observed decrease in the LFP gains well (*Figure 8d*, $r^2 = 0.84$). In addition, response lags of this model with respect to the stimulus increased with the mean stimulus (*Figure 8e*), similar to the slowdown observed in the LFP responses (cf. *Figure 7*). Finally, this model can also reproduce LFP responses to naturalistic, intermittent signals, approximating well the time trace (*Figure 8f,h*) and the dependence on previous whiffs (*Figure 8g*, compare to *Figure 2*).

Since the spiking machinery compensates for the slowdown in LFP responses to preserve the timing of odorant encounters, we wondered if a simplification of this model that ignores the slowdown of the LFP kinetics upon adaptation could be used to predict firing rate: $R_F = N(K_F \otimes \bar{a}(S, \varepsilon))$ where $N$ is a static nonlinearity, $K_F$ is a partially derivative-taking linear filter ($\otimes$ indicates convolution), and $\bar{a}(S, \varepsilon)$ the steady state solution of *Equation (1)* with $\varepsilon$ obeying *Equation 2*. This simplification reduces this model to a type of adaptive nonlinear-linear-nonlinear (NLN) model, which preserves Weber-Fechner Law and reproduces the firing rates of ORN in response to both naturalistic and Gaussian Stimuli (*Figure 8—figure supplement 2*). Thus it could be a useful tool in modeling ORN responses received by PNs, or in constructing computational models of the antennal lobe (*Assisi et al., 2011*; *Bazhenov et al., 2001*; *Berck et al., 2016*; *Capurro et al., 2012*; *Chong et al., 2012*; *Hopfield, 1991*; *Kee et al., 2015*; *Koulakov et al., 2007*; *Luo et al., 2010*; *Sanda et al., 2016*; *Satoh et al., 2010*; *Stevens, 2015*). The model reproduced the change in the input-output curves on increasing the mean stimulus (*Figure 8—figure supplement 2a*, cf. *Figure 3e*) and decreased gain inversely with the mean stimulus, consistent with the Weber-Fechner Law (*Figure 8—figure supplement 2b*). The model reproduced the observed decrease in the ORN gains (*Figure 8—figure supplement 2c*, $r^2 = 0.85$), and responses to naturalistic stimuli (*Figure 8—figure supplement 2d–f*).

## Discussion

We examined how ORNs encode naturalistic odor signals and characterized how ORN gain is dynamically modulated in response to stimuli. By using precisely controlled, repeatable odorant stimuli, and linear modelling, we found that: (1) ORN gain varies dynamically during responses to naturalistic stimuli, suppressing responses to whiffs following earlier whiffs (*Figures 1* and *2*). (2) Gain varies inversely with the mean stimulus (Weber-Fechner Law) and decreases with increasing stimulus variance (*Figures 2* and *3*). (3) Variance gain control was distributed across transduction and spiking, while mean gain control occurs only at transduction (*Figures 5* and *6*). (4) While gain control slows stimulus-to-transduction kinetics with increasing background intensity, this is compensated for by a corresponding speed up of transduction-to-spiking kinetics, which maintains the stimulus-to-firing rate kinetics relatively independent of stimulus intensity (*Figure 7*). A two-state model of the Or-Orco complex activity supplemented with an adaptive architecture similar to that of the classic bacterial chemotaxis system is sufficient to reproduce several key features of LFP response, firing rate response, and gain control (*Figure 8*).

### The Weber-Fechner Law in olfaction

The Weber-Fechner Law has been observed in several sensory systems, including vision (*Burkhardt, 1994*; *Laughlin and Hardie, 1978*; *Nikonov et al., 2006*), audition (*Riesz, 1928*), and somatosensation (*Holway and Pratt, 1936*). In olfaction, the Weber-Fechner Law was demonstrated at the LFP level (*Cafaro, 2016*; *Cao et al., 2016*). Here we directly measured ORN firing rate and stimulus intensity and found that the ORN firing rate exhibited Weber-Fechner gain scaling relative to the mean stimulus intensity for five different odor-receptor combinations (*Figure 3*, *Figure 3—*

*figure supplement 1*). These data suggest that olfaction shares the Weber-Fechner Law with other sensory systems.

What is the purpose of front-end Weber-Fechner gain scaling? ORNs are capable of spiking up to ~300 Hz (*Hallem and Carlson, 2006*); however, we found that with their compressive gain control, ORNs maintained firing rates between 0–50 Hz to fluctuating odor stimuli, even with a ten-fold increase in the mean stimulus. The ORNs' postsynaptic partners, the projection neurons (PNs) (*Olsen et al., 2010*), are most sensitive to ORN firing rates of below ~50 Hz (*Jeanne and Wilson, 2015*). Thus, gain scaling at ORNs could act to maintain ORN firing rates in the range that PNs are most sensitive to for a wide range of concentrations, avoiding saturation of the ORN-PN synapse.

## Variance gain control in olfaction

In principle, gain control in sensory systems could be affected by several moments of the stimulus distribution, measured over many timescales. In the visual system, gain control depends on stimulus mean and variance, and some studies have shown little dependence on higher moments like the skew and the kurtosis (*Bonin et al., 2006*; *Tkačik et al., 2014*). Cell-intrinsic variance gain control exists in a variety of systems, including the retina (*Beaudoin and Manookin, 2008*; *Zaghloul et al., 2005*), lateral geniculate nucleus (*Lesica et al., 2007*), auditory neurons(*Nagel and Doupe, 2006*), and cortex (*Díaz-Quesada and Maravall, 2008*; *Ringach and Malone, 2007*). Photoreceptors do not exhibit variance gain control, and variance adaptation arises only in the subsequent processing in bipolar cells and ganglion cells (*Baccus and Meister, 2002*; *Kim and Rieke, 2001*; *Rieke, 2001*).

What could be the functional role of variance gain control in olfaction? One possibility is to help ensure that ORN responses occupy a large fraction of their dynamic range (*Laughlin, 1981*). While we quantified our stimulus in terms of the first and second moments of the stimulus statistics in this study, these moments may not map simply onto the salient features that are most relevant to the fly's encoding scheme. Variance gain control could therefore be a consequence of an adaptive representation that is important to the coding properties of the ORN, but remains unknown to us. Nonetheless, because variance gain control is distributed between transduction and spiking machinery (*Figure 5*), and mean gain control slows down transduction (*Figure 7*) (*Cao et al., 2016*; *Nagel and Wilson, 2011*) but variance gain control does not (*Figure 7—figure supplement 1*), adaptation to the stimulus variance is mechanistically distinct from adaptation to stimulus mean.

## Models and mechanisms of ORN response and gain control

The results presented here, and the models that reproduce them, focus on the phenomenology of gain control in ORNs. Do these phenomenological results constrain possible mechanisms that could implement gain control in ORNs? Weber-Fechner gain scaling (*Figure 3*) can be reproduced by models using feed-forward loops (*Clark et al., 2013*; *Goentoro et al., 2009*), integral feedback (*Yi et al., 2000*), or both (*Schulze et al., 2015*). A detailed biophysical model of odor-receptor binding and channel opening has been proposed to account for transduction responses to odors (*Nagel and Wilson, 2011*). While this model can change gain via a negative feedback mechanism, it does not reproduce the Weber-Fechner law, or the slowdown of LFP kinetics upon adaptation. Here we showed that these features emerge if we assume that: (i) the activity $a$ of the Or-Orco complex feeds back onto the free energy difference $\varepsilon$ between the active and inactive state of the unbound Or-Orco complex, which in turns affects both the rates of activation and inactivation of the complex; and (ii) the rate at which $\varepsilon$ is modified only depends on the activity $a$. In summary, the steady state activity in our model depends nonlinearly on the stimulus, reproducing the effects of saturation in responses to naturalistic stimuli. Adaptation shifts the effective half-maximum of the input nonlinearity to the right, recapitulating Weber-Fechner gain control; and decreases transition rates from active to inactive receptor complexes, reproducing slowing LFP responses with adaptation.

Such an architecture reproduces the Weber-Fechner law and is similar to that of the bacterial chemotaxis system (*Barkai and Leibler, 1997*). There, adaptation is mediated by two antagonistic factors, one that acts on inactive complexes only, and another one that acts on active complexes (*Barkai and Leibler, 1997*). While the molecular architecture of the signaling pathway in ORNs has not been fully characterized, several studies have implicated calcium as a slow diffusible factor that could mediate adaptation to the mean stimulus (*Deshpande et al., 2000*; *Störtkuhl et al., 1999*). Decreasing extracellular calcium levels, or internal free calcium, breaks Weber-Fechner gain scaling

at transduction (*Cao et al., 2016*). Other mechanisms have also been implicated in adaptation of ORNs, like autoregulation of Orco via cAMP signaling (*Getahun et al., 2013*). While slower adaptive processes also exist, our data on responses to naturalistic stimuli (*Figures 1* and *2*), and data from paired-pulse experiments (*Cao et al., 2016*), suggest that some adaptation mechanisms act on fast timescales of several hundred milliseconds.

Many models that decrease gain with increasing mean stimulus also speed up response kinetics (*Clark et al., 2013*; *De Palo et al., 2012*; *Nagel and Wilson, 2011*; *Schulze et al., 2015*; *Seung, 2003*), describing well the phenomenology of other sensory systems where gain and response speed trade off (*Baylor and Hodgkin, 1974*; *Dunn et al., 2007*; *Nagel and Doupe, 2006*; *Payne and Howard, 1981*). However, in olfactory systems, transduction kinetics slow down with increasing stimulus background, both in insect ORNs (*Figure 7*, (*Cao et al., 2016*; *Nagel and Wilson, 2011*)) and in vertebrate ORNs (*Reisert and Matthews, 1999*). It is not trivial to devise systems in which kinetics slow down with increasing stimulus background. To exhibit this property, a system must increase its effective timescale of response with stimulus intensity, for example, by decreasing all reaction rates uniformly. Interestingly, our model also exhibits a slowdown in the LFP kinetics upon adaptation. This feature emerges intrinsically from the model architecture because: (i) the feedback of the activity onto the free energy difference $\varepsilon$ affects *both* the activation and deactivation rates of the complex ($w^+$ and $w^-$); and (ii) the Weber-Fechner gain control causes $\varepsilon$ to scale logarithmically with the stimulus, which in turn causes $w^+$ and $w^-$ to decrease.

Earlier work modelled the transformation from LFP to firing rates using a derivative-taking kernel (*Nagel and Wilson, 2011*). Here, we show that the temporal structure of these kernels depends on the adaptation state of the ORN, and must take derivatives on shorter timescales at higher stimuli to compensate for slowing transduction kinetics. Consistent with this, we see that the latency of spiking decreases increasing optogenetic drive (*Figure 7h*). Under these conditions, the change in spiking latency is smaller than the increase in transduction lags we observe (*Figure 7d–g*). While the mechanism of the speed up in spiking with increasing odor stimulus is not known, the neuron's ability to spike with shorter latencies relative to transduction could depend on the adapted state of its receptors, the level of intracellular calcium, or the distance of its membrane potential from firing thresholds.

What cellular mechanisms could give rise to gain control that is variance dependent? (*Figure 4*). We found that both the transduction machinery and the spiking machinery of the ORNs exhibit variance-sensitive gain-control (*Figures 5* and *6*). Variance gain control after transduction could arise from the spike generating machinery. Hodgkin-Huxley (HH) model neurons exhibit variance gain control (*Hong et al., 2008*; *Lundstrom et al., 2008*; *Yu and Lee, 2003*). Simpler neuron models, like the FitzHugh-Nagamo model (*Hong et al., 2007*), and the linear integrate-and-fire (LIF) model (*Yu and Lee, 2003*) also exhibit variance-dependent gain control. In the visual system, non-spiking bipolar neurons show variance gain control (*Baccus and Meister, 2002*; *Rieke, 2001*) so mechanisms for variance gain control in the absence of spike generation might be similar between these systems.

## Dynamic gain control could aid in naturalistic odor detection

Previous studies of olfactory adaptation employed conditioning and probe stimuli (*Cafaro, 2016*; *Martelli et al., 2013*; *Nagel and Wilson, 2011*), which typically adapt neurons over many seconds or minutes before testing response properties with a short probe. Other studies using paired pulse protocols (*Cao et al., 2016*) found that responses to brief pulses of odorant reduced gain on timescales as brief as 500 ms, which is close to the timescale of the neural response to odors (*Kim et al., 2011*; *Martelli et al., 2013*; *Nagel and Wilson, 2011*). Similar fast timescales of gain control have been observed in the visual system (*Burns et al., 2002*; *Baylor and Hodgkin, 1974*). We found that this fast gain control was employed by ORNs to dynamically control gain during responses to naturalistic odorant stimuli (*Figures 1–2*).

Dynamic gain control allows ORNs to respond to the rapidly changing statistics of natural odor plumes, letting gain decrease quickly in response to a large whiff and then ramp up again to a subsequent small whiff. Dynamic inhibition in the antennal lobe (*Nagel et al., 2015*; *Raccuglia et al., 2016*) would permit PNs to remain sensitive to these rapid changes in ORN firing rate, ensuring propagation of information about odor encounters to the brain.

### Invariant firing rate kinetics could improve odor-guided flight behavior

Insects follow odor plumes to their source to find food or reproductive mates (*Murlis et al., 1992*). For flies, this task is challenging since they fly fast (~30 cm/s) (*Tammero and Dickinson, 2002*) and odor filaments are narrow (*Murlis et al., 1992*). Even for relatively broad and static odor plumes, flies are within odor plumes so briefly that they experience plume contact and plume loss in quick succession (10–250 ms) (*van Breugel and Dickinson, 2014*). Olfactory search behavior in this setting consists of rapid flight surges on encountering odor plumes, and stereotyped crosswind casts on losing odor plumes (*van Breugel and Dickinson, 2014*). Navigation based on odor intensities alone may not be possible, as odor intensities are not informative about the direction to the odor source at length scales longer than 10 cm (*Murlis et al., 1992*). Indeed, there is a growing body of evidence underlining the importance of timing in olfaction (*Martelli et al., 2013*; *Rebello et al., 2014*; *Shusterman et al., 2011*; *Smear et al., 2013*, *2011*).

In this context, it may be important for the fly to know precisely *when* it encountered an odor filament. Previous studies have shown that kinetics of transduction slowed during adaptation (*Cao et al., 2016*; *Nagel and Wilson, 2011*), but kinetics of firing rate did not (*Martelli et al., 2013*). Here we reproduced both findings and resolved this apparent contradiction. We discovered that the spiking machinery speeds the kinetics back up. These complementary kinetic mechanisms mean that the timing of short odorant encounters is preserved in neural encoding, regardless of intensity. Such an encoding scheme could aid insects in navigating odor plumes to their source.

When a system responds identically, in amplitude and in kinetics, to stimuli that are different only in scale, the system is said to show fold change detection (FCD)(*Goentoro et al., 2009*). FCD thus implies the Weber-Fechner law, but systems can obey the Weber-Fechner Law without showing FCD. Another requirement for FCD is that response kinetics remain invariant with respect to the mean stimulus intensity. Thus, ORN responses are intriguingly similar to the response phenomenology of FCD networks (*Goentoro and Kirschner, 2009*).

Interestingly, olfactory adaptation is linked to flight in insects. Olfactory receptors (ORs) adapt and have co-evolved with flight (*Edwards, 1997*; *Getahun et al., 2013*; *Jones et al., 2005*), and occur only in flying insects (*Missbach et al., 2014*). In contrast, the more ancient ionotropic receptors (*Missbach et al., 2014*), found in all insects, do not appear to adapt to prolonged odor stimuli (*Cao et al., 2016*). While ORs play an important role in larval olfactory navigation (*Hernandez-Nunez et al., 2015*; *Mathew et al., 2013*; *Schulze et al., 2015*; *Gepner et al., 2015*), the statistics of odor signals close to surfaces, and in the air, where flying insects encounter them, may be very different (*Murlis et al., 1992*) (*Martelli et al., 2013*). Receptors capable of fast adaptation may allow flying insects to detect brief whiffs of airborne odors.

## Materials and methods

### Electrophysiology

#### Single sensillum recordings

Single sensillum recordings from *Drosophila* antennae were performed as described previously (*de Bruyne et al., 2001*; *Martelli et al., 2013*). The recording electrode was inserted into a sensillum on the antenna of an immobilized *Drosophila melanogaster* and a reference electrode was placed in the eye. Electrical signals were amplified using an Iso-DAM amplifier (World Precision Instruments). The ab3 sensillum was identified by (1) its size and location on the antenna (2) test pulses of 2-heptanone, to which the B neuron is very sensitive, (3) spike shapes (A spikes are larger than B spikes) and (4) spontaneous firing (ab3B fires at a higher rate than ab3A). Other sensilla were identified using test odors to which either the A or B neuron strongly responded to.

#### Spike sorting

All sensilla recorded from in this study contained two neurons (*de Bruyne et al., 1999*; *Song et al., 2012*). Generally, spikes from the 'A' neuron are larger than spikes from the 'B' neuron. However, spike amplitude and spike shape changed in our experiments with strong odor or light drive, due to a phenomenon called 'pinching' (*Olsson et al., 2006*), and due to small movements of the recording electrode relative to the sensillum. To identify spikes from the A neuron under these challenging

conditions, we developed a spike-sorting software package written in MATLAB (Mathworks, Inc.), available at https://github.com/emonetlab/spikesort. A copy is archived at https://github.com/elifes-ciences-publications/spikesort. This package uses the full spike shape, with various dimensionality reduction and clustering methods to reliably identify spikes from noise, and to sort identified spikes. This spike-sorting package performed with 99.5% accuracy compared to manually sorted data on a test dataset.

## Local field potentials

The local field potential was recorded by lowering the gain of the amplifier and switching to DC mode, where we recorded the sensillar potential without any low frequency cutoff. Spike detection and sorting was reliable in either mode. Since we were only interested in the deflections of LFP in response to a fluctuating odor stimulus, we band-passed the raw voltage in software to remove spikes and slow fluctuations.

## Fly stocks and genetic strategies

Flies were reared at 25°C on conventional fly medium (*Helfand and Carlson, 1989*). All experiments were performed on adult female flies 3–5 days post-eclosion. Unless otherwise mentioned, recordings were from ab3A neurons in Canton-S flies. In *Figure 6* and *Figure 7h*, we recorded from ab3A ORNs in w; Or22a-GAL4/+; UAS-Chrimson/+ flies. In these flies, only ab3A ORNs were sensitive to light, while ab3B neurons and nearby ab2 sensilla were not.

## Stimulus measurement

We used a Photo-Ionization Detector (PID) (200B, Aurora Scientific) to measure the odor stimulus during every experiment. Stimulus measurements occurred simultaneously with all electrophysiology, and the tip of the PID probe was <1 cm of the odor delivery tube and the fly (*Figure 1—figure supplement 1*). The PID was calibrated by depleting known volumes of pure odorants, and the response of the PID was found to be approximately linear with odorant flux (*Figure 1—figure supplement 1*). However, due to gradual changes in the sensitivity of the PID detector, odor intensity measurements are not comparable across experiments.

We measured the intensity of red light that we used to activate Chrimson at the location of the fly using a PM160 light power meter (Thorlabs). We used this to construct a function mapping control signals to our LED to light power in μW, and transformed control signals into light power using this function.

## Odor stimulus generation

### General principle

Odorants in gas phase were delivered to the antenna by blowing air over pure monomolecular odorants in liquid phase. The flow rate of air over the liquid odorant determined the gas phase concentration.

### Controlling air flows

Mass Flow Controllers (MFCs) (Aalborg instruments and Controls, Inc. and Alicat Scientific) were used to regulate airflows. Dynamic response parameters of Alicat MFCs were chosen either for high speed, and driven with switching times of up to 20 ms (at the cost of reproducibility) or were chosen for high precision, and driven with switching times of 100 ms (at the cost of very fast stimulus control). An odorized airstream (0–200 mL/min) was fed into a main airstream (2 L/min) that was delivered through a glass tube positioned within 10 mm of the fly's antenna. The secondary airstream passed through a scintillation vial with a machined plastic screw-top lid containing pure odorant (*Figure 1—figure supplement 1*). Using pure odorant and gas phase dilution permitted excellent reproducibility of the odor stimulus. All tubing was made of chemical-resistant PTFE tubing (McMaster Carr, stock #5239K24).

By varying the control signals to the MFC bank, steps, pulses, and frozen noise waveforms with arbitrary distributions could be reliably delivered. We wrote a general-purpose acquisition and control system called kontroller (available at https://github.com/emonetlab/kontroller) in MATLAB

(Mathworks, Inc.) to control MFCs, valves and LEDs and to collect data from electrophysiology and the stimulus measurement.

## Naturalistic stimulus (*Figure 1*)

To generate naturalistic odor stimuli, we randomly varied flow rates over 0–200 mL/min, and used a small solenoid valve (Lee Co.) to deliver 50 ms whiffs of odorant. We used the same frozen random sequence in subsequent trials.

## Stimulus with changing mean (*Figures 2* and *4*)

To generate approximately Gaussian-distributed stimuli that differed in their mean, but with similar variances, we started with the ansatz that air flow rates were proportional to measured gas-phase stimulus. We then defined target Gaussian distributions that differed only in their mean, and a parametric distribution from which we drew control signals to the MFC. Using kontroller to automate the process, we performed a direct search on hardware to find the best distribution of control signals that was closest to the desired Gaussian distribution. Further rounds of off-line numerical optimization using nonparametric models of the delivery system ensured that the resultant stimulus distributions were as close to Gaussians and with variances as similar to one another as possible.

## Stimulus with changing variance (*Figures 3–4*)

We used two MFCs driven by control signals with different variances to generate two Gaussian-distributed stimuli with differing variances. Solenoid valves (Lee Co.) were used to switch from one airstream to the other every 5 s. Control signals were iteratively optimized using simulations and kontroller till the mean stimulus intensity from the two odor lines was indistinguishable.

## Numerical methods

### Statistics of naturalistic odor

Since the intensity distribution in our naturalistic odor stimulus was very broad, we defined whiffs of odor as short excursions of the odorant signal above the noise floor. Blanks were defined as the periods of time between whiffs. Whiff intensities were broadly distributed, and were fit with a functional form proposed in *Celani et al. (2014)*. Both whiff and blank duration distributions were fit with a power law with exponent $-3/2$, following theoretical calculations for a jet flow (*Celani et al., 2014*) (*Figure 1—figure supplement 3*). Mean and variance of naturalistic odor stimuli were computed in non-overlapping windows of length $\tau$ (*Figure 1—figure supplement 3e*, $\tau = 400$ ms). Window lengths were varied from $\tau = 10$ ms to $\tau = 10$ s (*Figure 1—figure supplement 3f*).

### Estimating deviations in response to naturalistic stimulus

For every whiff shown in *Figure 1e–f*, we computed the median response for all whiffs in a bin centered around that whiff's stimulus amplitude, that encompassed other whiffs if their amplitude was within 10%. The fractional deviation in response to a given whiff $i$ is defined as $D_i = \left( R_i - \tilde{R} \right) / \tilde{R}$, and is a dimensionless number that is negative when responses are smaller than the median ($\tilde{R}$). To determine if the time to the previous whiff and the amplitude of the previous whiff of negative deviations and positive deviations were different, we used a 2-sample two-dimensional Kolmogorov-Smirnoff test, based on the method proposed by Peacock (*Peacock, 1983*).

### Filter extraction

Linear filters can be reliably estimated even in the presence of a output nonlinearity for white Gaussian inputs (*Chichilnisky, 2001*). We extracted linear filters from measured Gaussian odor stimuli and ORN responses using least-squares fitting. Given time series of input $S$ and response $R$, we reshaped $S$ into a matrix $\hat{S}$ where each row of $\hat{S}$ contained the stimulus up to $N$ samples in the past where $N$ is the length of the filter to be calculated. Using this matrix, we computed the stimulus covariance matrix $C = S^T S$. The linear kernel that is the best linear predictor of $R$ given $S$ is given by $K = C \backslash \hat{S}^T R$. However, since input signals have autocorrelation functions with power approaching 0 at high frequencies, estimated filters were occasionally dominated by high-frequency artifacts. To remove these, we regularized $C$, using $\hat{C} = C + rI$ where $I$ is the identity matrix and $r$ is a regularization factor

in units of the mean eigenvalue of $C$. Finally, we normalized the filters as described in (*Baccus and Meister, 2002*) to preserve the units of the stimulus in the linear projections. The regularization value $r$ was chosen by cross-validation to create filters that best predicted the data.

Linear filters extracted from naturalistic stimuli (*Figure 1*, *Figure 1—figure supplement 4*) are not unbiased estimates of the true filter, since the stimulus is not Gaussian. Similarly, if a front-end nonlinearity precedes a linear filtering step in a system, this fitting procedure will not yield an unbiased estimate of the true filter. Nonetheless, linear filters computed by least squares fitting are the filters that best predict $R$ given $S$, in the least-squares sense.

## ORN input output curves (solid lines in *Figures 3e* and *4e* etc.)

We defined ORN input-output curves to be the output nonlinearity of a LN model, which were estimated by plotting ORN response *vs.* the projected stimulus, and then computing a piecewise linear function using 50 bins along the horizontal axis. Computing piecewise linear functions allowed us to visualize the output nonlinearity without making explicit assumptions of the functional form of the nonlinearity. Dashed lines in *Figure 4e* are the cumulative distributions of the stimulus, computed over all the data.

## Estimation of gain

In general, for any system with a single stimulus and response, we define the gain of the system by measuring the slope of the nonlinearity in the best fit LN model, normalizing the filters to preserve the scale of the stimulus as in *Baccus and Meister (2002)*. With Gaussian stimuli that only gently perturb the system, the nonlinearity is simply a straight line, and the gain is computed by the average slope of a linear fit to the output nonlinearity (*Figure 3*). When output nonlinearities are strong, we estimated gain by the slope at the midpoint of the nonlinearity (*Figures 4* and *5*). We measured three different gains: (1) transduction gains, from the stimulus to the LFP, (2) firing gains, from the LFP to the firing rate, and (3) overall ORN gains, from the stimulus to the firing rate. Transduction gain had units of (mV/V) since deflections in LFP are measured in mV and the stimulus is measured in V. Similarly, firing gain had units of Hz/mV. Overall ORN gain had units of Hz/V when stimulating ORNs with fluctuating odor, and had units of Hz/µW when stimulating with light.

In experiments where we changed the variance (*Figure 4*), low-variance epochs tended to have a mean stimulus ~8% higher than high-variance epochs (*Figure 4—figure supplement 1*), despite our best efforts to keep the stimulus mean the same. To estimate gain changes solely due to the change in stimulus variance, we divided the projected stimulus by the mean stimulus in each epoch in each trial (*Figure 4e*). Differences in gain between high- and low-variance epochs remain significant even without this correction (*Figure 4—figure supplement 1b–c*). A single filter was used to project stimuli in both low- and high-variance epochs; changes in gain from low- to high-variance stimuli thus appear solely in the nonlinearity (*Figure 4—figure supplement 1d*).

## Measuring lags

In all our data, we measured the stimulus together with the response of ORNs. This allowed us to estimate transduction and firing lags with respect to the stimulus. In general, we estimated response lags by computing cross-correlation functions from the stimulus to the response. Lag was defined to be the location of the peak of the cross-correlation function. (*Figure 7a–b*).

## Estimation of variance gain control timescale

To estimate the timescale of variance gain control (*Figure 4*), we computed input-output curves from the projected stimulus to the firing rate in 50 ms bins, pooling all trials together. This allowed us to estimate gain in 50 ms bins, together with the stimulus contrast (standard deviation/mean). We plotted time series of instantaneous gain and stimulus contrast (*Figure 4h*), and observed that while the stimulus contrast changed rapidly after the switch from high to low variance (at t = 5 s), the instantaneous gain changed more slowly, but still changed in ~130 ms, suggesting that timescale of variance gain control is relatively rapid.

## Statistical tests

To determine if ab3A transduction-to-firing gain varied with the mean stimulus (*Figure 5d*), we used a Spearman rank correlation test. To determine if gains varied significantly from low to high variance epochs (*Figures 4* and *5*), we first reshaped the raw data into trials 10 s long. Each trial consisted of a high variance epoch followed by a low variance epoch. Each trial was fit with three linear models, and yielded three pairs of gains, for transduction gain, firing gain, and total ORN gain. We discarded all trials where any linear model fit was poorly correlated with data, retaining only trials where all fits had high correlations with data ($r^2 > 0.8$, see *Figure 4—figure supplement 1f–g*). We used the Wilcoxon signed rank test on these tuples to determine if the difference in gains in the low and high variance epochs was statistically significant.

## Modelling

### Stimulus binding and the activity of the Or-Orco complexes

We assume that Or and Orco form a complex that can exist in two conformations that can bind ligand. The concentration of unbound active complexes is $C^*$ and that of unbound inactive one is $C$. The corresponding concentrations for the bound complexes are $C^*S$ and $CS$ (*Figure 8a*). The fraction of active Or-Orco complexes is therefore (*Figure 8a*):

$$a = \frac{C^* + C^*S}{C + CS + C^* + C^*S}$$

(Un)binding of odorant is taken to be much faster than the (in)activation. Thus, the probability to be bound in the active and inactive cases are

$$P_b = \frac{S}{S + K_{off}}$$

$$P_b^* = \frac{S}{S + K_{on}}$$

Here $K_{on} < K_{off}$ are the dissociation constants for each state. Let the free energy difference in units of $k_B T$ between the active and inactive states be $\varepsilon$ and $\varepsilon^{\mathrm{b}}$ when unbound and when bound, respectively. For simplicity, we assume detailed balance (this can easily be relaxed (*Skoge et al., 2013*)), which constrains the free energy difference between $C^*S$ and $CS$ to be $\varepsilon^{\mathrm{b}} = \varepsilon + log(K_{on}/K_{off})$. The activation kinetics can then be described by

$$\frac{dC^*}{dt} = w_+^u \, C - w_-^u \, C^*$$

$$\frac{dC^*S}{dt} = w_+^b \, CS - w_-^b \, C^*S$$

where the rates are

$$w_\pm^u = \frac{\alpha}{1 + e^{\pm\varepsilon}}$$

$$w_\pm^b = \frac{\alpha}{1 + e^{\pm\varepsilon^{\mathrm{b}}}}.$$

We constrained the energy barrier between the active and inactive conformations by making the simplifying assumption that $w_+^u + w_-^u = \alpha = w_+^b + w_-^b$, where $\alpha$ is an intrinsic switching rate (see e.g. *Skoge et al. (2013)*). From these considerations we can then derive *Equation (1)* from the main text, which describes the dynamics of the activity:

$$\frac{da}{dt} = (1 - a) \, w_+(S, \varepsilon) - a \, w_-(S, \varepsilon)$$

with the rates

$$w_+(S,\varepsilon) = P_b w_+^b + (1 - P_b)w_+^u = \frac{S}{S + K_{off}} \frac{\alpha}{1 + e^{\varepsilon + log\left(\frac{K_{on}}{K_{off}}\right)}} + \frac{K_{off}}{S + K_{off}} \frac{\alpha}{1 + e^\varepsilon} \tag{3}$$

$$w_-(S,\varepsilon) = P_b^* w_-^b + \left(1 - P_b^*\right)w_-^u = \frac{S}{S + K_{on}} \frac{\alpha}{1 + e^{-\left(\varepsilon + log\left(\frac{K_{on}}{K_{off}}\right)\right)}} + \frac{K_{on}}{S + K_{on}} \frac{\alpha}{1 + e^{-\varepsilon}} \tag{4}$$

Given a steady state signal $S$, the activity relaxes towards

$$\bar{a}(S,\varepsilon) = \frac{1}{1 + \frac{w_-}{w_+}} = \frac{1}{1 + e^\varepsilon \frac{1 + S/K_{off}}{1 + S/K_{on}}}.$$

Where the overbar indicates that *Equation (1)* is solved at steady state. We model adaptation by supposing that the activity feeds back onto the free energy difference $\varepsilon$ with rates that depend only on the activity (the effective switching rate $\alpha$ is constant):

$$\frac{d\varepsilon}{dt} = \beta(a - a_0)$$

where $\beta$ is the rate of adaptation. Note that the free energy is bounded both from below and from above. In practice, we only need the lower bound $\varepsilon_L < \varepsilon$. At steady state (for values of $S$ high enough that $\varepsilon_L < \varepsilon$) we have $\bar{a} = a_0$ which implies that

$$\bar{\varepsilon}(S) = log\left(\frac{1 - a_0}{a_0} \frac{1 + S/K_{on}}{1 + S/K_{off}}\right)$$

Thus, at steady state, adaptation causes the free energy difference of the complex to increase with the logarithm of the background signal intensity.

## Kinetic slowdown upon adaptation

Substituting $\bar{\varepsilon}(S)$ into the definitions of the (in)activation rates, we get

$$\bar{w}_+(S) = \frac{S}{S + K_{off}} \frac{\alpha}{1 + \frac{K_{on}}{K_{off}} \frac{1 - a_0}{a_0} \frac{1 + S/K_{on}}{1 + S/K_{off}}} + \frac{K_{off}}{S + K_{off}} \frac{\alpha}{1 + \frac{1 - a_0}{a_0} \frac{1 + S/K_{on}}{1 + S/K_{off}}}$$

$$\bar{w}_-(S) = \bar{w}_+(S) \frac{1 - a_0}{a_0}.$$

When $K_{on} < K_{off}$, the rates $\bar{w}_+(S)$ and $\bar{w}_-(S)$ are decreasing function of $S$ in the range

$$0 \le S \le \sqrt{\frac{a_0(1 - K_{on}/K_{off}) - K_{on}/K_{off}}{1 - a_0(1 - K_{on}/K_{off})} K_{on} K_{off}} \approx \sqrt{\frac{a_0}{1 - a_0} K_{on} K_{off}}$$

where the approximation is valid when $K_{on} \ll K_{off}$. In our case this bound on $S$ is large (~40 V in PID measurement units) and in our experiments the rates are decreasing functions of $S$ over the entire range.

## Receptor activity to LFP

The output of the model described above is a time series of the fraction of receptors that are active, $a(t)$. Receptor activation can lead to the opening of other channels, which results in a transduction current that we measure as changes in the LFP. To generate LFP responses from this, we use

$$R_{LFP} = C_0(K_{LFP} \otimes a(t)) \tag{5}$$

where $\otimes$ represents a convolution and $K_{LFP}$ is a linear time-invariant mono-lobed filter that is given by:

$$K_{LFP} = \frac{t^m e^{\frac{-t}{\tau}}}{m! \tau^{m+1}} \theta(t)$$

## Receptor activation to firing rates

Since firing rates cannot be negative, and since the LFP to spiking transformation has been shown to be partly differentiating (*Nagel and Wilson, 2011*), we generated firing rate responses from the receptor activity using

$$R_F = N(K_F \otimes \bar{a}(t))$$

where $N$ is an output nonlinearity which is a simple threshold linear function $(N(x < 0) = 0); N(x > 0) = Cx)$. $\otimes$ represents a convolution and $K_F$ is a linear time-invariant filter that is given by the sum of two other kernels:

$$K_F = K_1 + \alpha K_2$$

where each kernel is parameterized by a Gamma function:

$$K_i = \frac{t^m e^{\frac{-t}{\tau_i}}}{m! \tau_i^{m+1}} \theta(t), \ i \in \{1, 2\}$$

## Code availability

1. spikesort. MATLAB toolbox to sort spikes from extracellular recordings. Available at https://github.com/emonetlab/spikesort.
2. kontroller. MATLAB toolbox to acquire data and run experiments on National Instruments hardware. Available at https://github.com/emonetlab/kontroller

## Acknowledgements

We gratefully acknowledge helpful conversations with John Carlson, Julijana Gjorgjieva, Charles Greer, Joe Howard, Carlotta Martelli, and Steven Zucker. We are thankful to John Carlson for giving us access to some of the instrumentation in his lab. MD, SG-S and TE were partially supported by the Whitehall Foundation. JL was supported by the Natural Sciences and Engineering Research Council of Canada (NSERC) Postgraduate Scholarships-Doctoral Program PGSD2-471587-2015. DAC was supported by a Sloan Research Fellowship, a Searle Scholar Award, and the Smith Family Foundation.

## Additional information

### Funding

| Funder | Grant reference number | Author |
|---|---|---|
| Whitehall Foundation | | Srinivas Gorur-Shandilya<br>Mahmut Demir<br>Thierry Emonet |
| Sloan Research Fellowship | | Damon A Clark |
| Searle Scholar Award | | Damon A Clark |
| Smith Family Foundation | | Damon A Clark |
| Natural Sciences and Engineering Research Council of Canada | PGSD2-471587-2015 | Junjiajia Long |

The funders had no role in study design, data collection and interpretation, or the decision to submit the work for publication.

## Author contributions

SG-S, Conceptualization, Data curation, Software, Formal analysis, Validation, Investigation, Visualization, Methodology, Writing—original draft, review and editing, designed the odor delivery system, wrote control software, did all the analysis; MD, Conceptualization, Investigation, Methodology, Writing—original draft, Performed electrophysiology measurements; JL, Formal analysis, Methodology, Writing—review and editing, built the mathematical model; DAC, Conceptualization, Formal analysis, Supervision, Validation, Investigation, Visualization, Methodology, Writing—original draft, review and editing; TE, Conceptualization, Formal analysis, Supervision, Funding acquisition, Validation, Investigation, Visualization, Methodology, Writing—original draft, Project administration, Writing—review and editing

## Author ORCIDs

Srinivas Gorur-Shandilya, http://orcid.org/0000-0002-7429-457X
Mahmut Demir, http://orcid.org/0000-0002-3278-7843
Junjiajia Long, http://orcid.org/0000-0003-3754-9614
Damon A Clark, http://orcid.org/0000-0001-8487-700X
Thierry Emonet, http://orcid.org/0000-0002-6746-6564

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
