## [Decision Letter]

[Editors’ note: a previous version of this study was rejected after peer review, but the authors submitted for reconsideration. The first decision letter after peer review is shown below.]

Thank you for submitting your work entitled "Olfactory receptor neurons use fast dynamic gain control to encode intermittent odorant stimuli" for consideration by *eLife*. Your article has been favorably evaluated by a Senior Editor and three reviewers, one of whom, Fred Rieke (Reviewer #1), is a member of our Board of Reviewing Editors. The following individual involved in review of your submission has agreed to reveal their identity: Tim Holy (Reviewer #2).

Our decision has been reached after consultation between the reviewers. Based on these discussions and the individual reviews below, we regret to inform you that your work will not be considered further for publication in *eLife*, at least without substantial revisions.

Two main points emerged in discussion among the three reviewers about the paper. We highlight these points because we felt they could change how the data is interpreted. All reviewers agreed that these points were important to deal with before we could consider the paper further. Individual reviews, attached below, will both elaborate on these points as well as detail other important issues to consider.

Adaptation vs. saturation.

Adaptation as defined in the paper includes both time-dependent changes in cellular processing as well as static nonlinearities in the response – such as receptor saturation. We felt it was quite important to separate these effects as much as possible. This could, for example, involve fitting NLN models to the responses, providing a "front-end" nonlinearity that could account for receptor saturation. Several related comments in the individual reviews involve how gain is defined and how much the conclusions depend on the definition of gain.

Validity of LN models.

The LN models the form the core analysis tool in the paper are not validated. Several assumptions are central to the analysis. Some of these points may also originate from a lack of clarity in the paper about details of how the LN models were fit. Most importantly, it is not clear in all cases how the linear filter used to project the stimulus was computed. For example, since the standard linear filter calculation assumes symmetry of the stimulus, it is not clear how filters were computed for the naturalistic odor stimuli.

*Reviewer #1:*

This paper describes adaptation to the mean and variance of odorant stimuli, with a particular focus on adaptation to natural inputs. The question is quite interesting, and for the most part I found the conclusions of the paper well supported by the data presented. Several issues, however, limit my enthusiasm.

How good are LN model fits to the data?

The paper relies extensively on LN models to analyze adaptation. More validation of this approach is needed – e.g. how much of the response variance do LN models describe? Related to this point – are the linear filters used in analysis of responses to naturalistic inputs derived from responses to Gaussian noise inputs (this seems to be the case from the Methods)? If not, how are the filters computed? If the filters are computed for Gaussian noise inputs, are they still valid for naturalistic inputs? Similarly, in Figure 4 – are the low variance and high variance filters assumed to be the same, and is this indeed correct?

LFP as a proxy for transduction.

The LFP is used to estimate properties of odor transduction. It is not clear from the paper how good a proxy the LFP is for transduction, specifically whether it is accurate enough to support the conclusions drawn. More verification of the validity of using the LFP is needed.

Gain definition.

The gain in Figure 2 is defined as the slope of the firing rate vs. projected stimulus curve. But doesn't that mean comparing different firing rates for different whiffs – and hence potentially confounding adaptive effects with static nonlinearities in the responses? This issue also comes up in Figure 3.

Compensation for slowing of transduction.

The conclusion that speeding of post-transduction processes with adaptation compensate for slowing of transduction is not well developed, especially for a major conclusion of the paper. This relates to the point above about using the LFP as a proxy for transduction.

*Reviewer #2:*

The submission by Gorur-Shandilya and colleagues examines the dynamics of odorants responses in *Drosophila* ORNs using both naturalistic (plume-based) stimuli and gaussian stimuli of controlled mean and variance. The authors show that adaptation occurs on rapid time scales in a manner consistent with the well-known Weber-Fechner "law." By measuring both field potentials and spiking, the authors attempt to dissociate transduction dynamics from spiking dynamics, and discover that the two have compensatory kinetic characteristics, so that overall the firing preserves the dynamics of the stimulus.

This study brings the analysis of *Drosophila* ORN dynamics closer to the standard set in, e.g., phototransduction and several other sensory systems, and for that reason has considerable value. Overall, the results suggest a fair degree of correspondence with the early stages of visual encoding, right down to the magnitude of the relevant time scales. As the authors acknowledge, it is primarily a "phenomenology" study, leaving most questions of mechanism for future work. (The approximate dissociation of spiking vs. transduction is something of an exception, however.) However, the history of other systems argues that quantitative characterization of the phenomenology is an essential step along the path to discovering the underlying machinery.

At a big-picture level, the Introduction sets the stage well, the manuscript as a whole appears scholarly, and the claims seem well-supported by the data and analyses. A key component of this work is the impressive control over stimulus delivery, which is much harder for olfactory stimuli than for visual stimuli, and the authors clearly went to great lengths to achieve this control. Consequently, this study has a lot to recommend it, and little to dislike. My concerns are minor, focusing on the clarity of descriptions, particularly in the Materials and methods section (which is not up to the standard set in the rest of the manuscript). In its current state I suspect that readers will have considerable difficulty understanding exactly what was done at several points in the manuscript. There are some places where I question details of the results, but it seems likely that this is due to a misunderstanding of the precise form of the analysis.

Figure 2: one oddity is that the tangent of the f-s curve even at 0 appears strongly correlated with the ultimate size of the excursion. Naively, this would appear to violate causality, but of course it could reflect correlations in the stimulus. Nevertheless, it begs the question of how much of this analysis is really dominated by the stimulus correlations; there don't appear to be *any* examples of a large stimulus that were not preceded by some adaptation, and that leads to concern about the thoroughness of this particular test and analysis.

*Reviewer #3:*

This paper examines the responses of *Drosophila* olfactory receptor neurons to dynamic fluctuating stimuli and analyzes their responses predominantly using linear filter methods. Based on these analyses the authors argue that ORNs exhibit fast adaptation that allows them to better encode the properties of odor plumes. The experiments appear to be carefully performed and the data appear to be solid. However, I have concerns about the novelty of the approach and findings, as well as some of the analyses and interpretations.

Linear filter methods have been used for many years to approximate the response of a sensory neuron to a complex time-varying stimuli. A common theme of these studies is the finding that the linear approximation of the response varies as a function of stimulus statistics such as the mean and variance (e.g. Fairhall 2001, Baccus and Meister 2002). In particular, the approximate gain of the neuron generally decreases with increased stimulus variance, a phenomenon known as "gain scaling" or "variance scaling". Following on these studies several groups asked whether single neurons could exhibit this property in response to current injection, and found that it could. Subsequent modeling studies have shown that Hodgkin-Huxley models and other spiking models are capable of producing gain scaling (e.g. Lundstrom et al., 2008). Therefore, the finding that *Drosophila* ORNs exhibit gain scaling, and that some of this scaling arises at the level of spike generation does not seem novel or surprising.

In the *Drosophila* olfactory neurons studied here, several papers have examined responses to temporally modulated stimuli (e.g. Kim 2010, Nagel 2011, Martelli 2013, Cafaro, 2016, Cao, 2016). Multiple studies have found that transduction response magnitude (measured intracellularly or by LFP) is well described by a Hill equation, (that is, it rises sigmoidally with log odor concentration), as one would predict from the basic biochemistry of receptor activation (Cao, Cafaro, Nagel). Second, they have found that OR-expressing ORNs exhibit calcium-dependent adaptation that dynamically adjusts sensitivity and gain (Cao, 2016). The approach taken here confounds these two phenomena, as it assumes that responses are a linear function of concentration (here shown to be linearly related to PID voltage). Therefore, the "adaptation" reported by the authors likely contains contributions from the nonlinearity inherent in receptor activation, as well as true dynamical adaptation. This might be addressed by computing filters based on the log concentration, rather than linear concentration, and by comparing "adaptation" (estimated from projections onto linear filters) in OR- and IR expressing ORNs, as IR-expressing ORNs do not exhibit calcium-dependent adaptation, but do show activation nonlinearities.

An issue in the paper is what is meant by "adaptation." The authors seem to argue that adaptation is any change in the linear approximation to the response. For example, in the Introduction they state that they are focusing on phenomenology, which they call "a prerequisite to understanding the mechanisms that implement them." In the Discussion, they say that "'adaptation state is a construction to understand nonlinearities in that response." However, many other parts of the paper imply that these "adaptations" are actively produced by the cell in order to optimize coding. For example, in the title "Olfactory receptor neurons use fast dynamic gain control", Results "small isolated whiffs were amplified…suggesting that ORNs adjusted their gain dynamically", "suggesting that ORNs actively changed their gain." The question of whether certain coding phenomena arise from active adaptation processes, or from unavoidable nonlinearities of receptor-ligand interactions seems germane to the framing of the paper, and the question of whether the phenomena they describe are specializations of insect neurons for encoding odor plumes or general properties of receptor-activated neurons.

A second issue is the characterization of phenomenology versus mechanism. Although the authors claim to focus on phenomenology, a significant part of the paper is devoted to the distinguishing phenomena arising at the level of transduction/LFP, and those arising at the level of spikes- an attempt to place phenomenology in a mechanistic context. Given that multiple stages have been experimentally identified in the transduction process (Cao et al. 2016), and that many models of spike generation are available, it does not seem fair to call this phenomenological characterization a "necessary prerequisite" to mechanistic understanding.

Other comments:

The finding that ORN spikes seem to counteract some of the slowing of transduction responses is interesting and novel but not well-developed.

I am concerned about the linear filter method applied to responses to sparse stimuli in Figure 2. As stated in the Materials and methods, linear filters can be reliably estimated in the presence of an output nonlinearity if the stimuli are Gaussian- however, this is generally not true for stimuli that are strongly skewed, such as the one used here.

In general the authors do not report the goodness of fit of their linear model, making it difficult to interpret whether the observed changes in gain (where gain is the slope of the projected versus real response) are significant.

The authors state: "It remains unclear whether the ORN's ultimate output- the firing rate- follows the same Weber-Fechner scaling." However, Cafaro, 2016 does address this question for ORN spikes, as well as PN membrane potential and spikes.

Introduction, last paragraph: It is unclear at this point in the text why the authors choose to focus on a history of 300ms. This should be better motivated.

Subsection “Fast gain control maintains timing information of naturalistic odor signals”, third paragraph. It would be helpful to at least briefly mention how lags were computed in the Results.

Subsection “Variance gain control in olfaction”. No potential mechanisms for gain scaling of transduction responses are given.

Subsection “Fast gain control could aid in naturalistic odor detection”. The authors state that paradigms that employ conditioning and probe stimuli cannot easily quantify the dynamics of gain change. This does not seem accurate. These paradigms explicitly allow the experimenter to distinguish the stimuli used to generate adaptation from those used to test it. Small combinations of pulses are used to examine the dynamics of gain change in transduction in Cao et al. 2016.

Subsection “Stimulus measurement”. It would be helpful to include a diagram of the odor delivery apparatus along with calibration data.

[Editors’ note: what now follows is the decision letter after the authors submitted for further consideration.]

Thank you for submitting your article "Olfactory receptor neurons use gain control and complementary kinetics to encode intermittent odorant stimuli" for consideration by *eLife*. Your article has been reviewed by two peer reviewers, and the evaluation has been overseen by a Reviewing Editor and a Senior Editor. The following individual involved in review of your submission has agreed to reveal their identity: Tim Holy (Reviewer #3).

The reviewers have discussed the reviews with one another all agreed that the revisions have improved the paper substantially. There are a few outstanding issues outlined in the reviews below.

*Reviewer #2:*

The revised manuscript describes encoding of dynamic stimuli by ORNs. The authors have addressed several of the major concerns cited by all reviewers. First, they find that a static front-end nonlinearity (a Hill function) can account for much of the nonlinearity observed in response to naturalistic odor stimuli, but that a dynamic adaptation component remains. Second, they have expanded their analysis of the dynamics of spiking and transduction, to show that speeding of spiking compensates for slowing of transduction with increased stimulus mean, resulting in spike timing that is independent of stimulus mean- an interesting and novel result. Finally, they have developed a receptor-based model to account for the observed forms of adaptation.

Overall the manuscript is much improved. I think it makes a substantial contribution to the literature. I do have some remaining issues with the manuscript that I think should be resolved prior to publication.

1) The Hill function as a source of nonlinearity.

The reviewers previously asked whether many of the nonlinearities described in the data could arise from receptor-ligand interactions, which should give rise to a nonlinear relationship between odor concentration and transduction. The authors have re-analyzed their data and found that responses to natural stimuli are well-described by such a relationship (Figure 1), although context-dependent deviations from this relationship remain (Figure 2). However, it seems that this understanding is not entirely integrated into the text. For example, the title of the first section is "ORN responses to naturalistic odorant stimuli show deviations from linearity that arise from adaptation and saturation" which implies that saturation is the only nonlinearity (not the logarithm implied by the Hill function). Much time is spent on showing that responses deviate from linearity, and then receptor-ligand interactions are suggested as an alternative to output nonlinearities, which seems odd, given that receptor-ligand nonlinearities are an obligate feature of a system composed of odorant receptors that bind and are activated by odorants. A finding that odor concentration is encoded linearly by such a system would be much more surprising. Along these lines, in the Introduction, the authors state that "ORNs employed front-end nonlinearities" which makes it sound like an active choice on the part of ORNs. I think a fairer statement of the findings would be that a front-end nonlinearity, combined with dynamic adaptation that shifts the midpoint of this nonlinearity, can account for ORN responses to a variety of dynamic stimuli.

2) Explanation of Weber-Fechner scaling.

Understanding the origin of Weber-Fechner scaling is a major goal of the paper. Along the lines above I would suggest that Figure 3—figure supplement 2 should be made part of the main text. This shows that the expected front-end nonlinearity, combined with adaptation that shifts the activation curve to the right, is sufficient to account for ORN responses to stimuli with different means. As such a rightward shift with adaptation has been previously reported (Kaissling, 1987, Nagel and Wilson, 2011), this makes for an elegant explanation of the observed responses.

The authors state that the receptor model alone cannot account for Weber-Fechner scaling (response to reviewers 1B). It is true that the receptor model alone does not produce responses that change gain and have similar mean, as in the data. However, it does seem that the receptor model alone (Figure 3—figure supplement 2, panel c) shows a decrease in gain with increased mean. If the output of this simple model were plotted as in Figure 3 what would it look like? Does the change in gain arise from the dynamic adaptation (the shift in half-max) or from the nonlinearity?

3) Receptor-based model.

The authors present a new model, based on a 2-state receptor model, to account for the observed adaptation. As I understand it, their model makes two assumptions: (1) receptor activation/inactivation are the rate limiting steps, and (2) receptor activity feeds back onto these rates, slowing both. The model thus makes specific predictions about what biophysical steps give rise to adaptation.

Given the findings in Figure 3—figure supplement 2, it seems like the main features necessary to explain the results are (1) steady-state output rises as S/S+K, (2) adaptation shifts K to the right, (3) adaptation decreases transition rates, leading to slower responses. The authors should clarify which aspects of their model are essential for reproducing the key features of the data.

4) Use of the LFP as a proxy for transduction.

It is true that it has become somewhat standard in the literature to do this but the LFP should be interpreted with a bit of caution. In Nagel and Wilson 2011, palp LFPs were used because LFPs in this structure were found to more closely reflect activity in a single sensillum (presumably because of the greater spacing between sensilla). A brief caveat noting that LFPs can contain a contribution from nearby sensilla would be welcome.

*Reviewer #3:*

The revisions thoroughly address my concerns. I am quite satisfied with the revised manuscript.

---

## [Author Response]

[Editors’ note: the author responses to the first round of peer review follow.]

*Two main points emerged in discussion among the three reviewers about the paper. We highlight these points because we felt they could change how the data is interpreted. All reviewers agreed that these points were important to deal with before we could consider the paper further. Individual reviews, attached below, will both elaborate on these points as well as detail other important issues to consider.*

Adaptation vs. saturation.

*Adaptation as defined in the paper includes both time-dependent changes in cellular processing as well as static nonlinearities in the response – such as receptor saturation. We felt it was quite important to separate these effects as much as possible. This could, for example, involve fitting NLN models to the responses, providing a "front-end" nonlinearity that could account for receptor saturation. Several related comments in the individual reviews involve how gain is defined and how much the conclusions depend on the definition of gain.*

We agree with the reviewers that our previous analysis of gain control did not distinguish between effects of adaptation with those of a potential front-end nonlinearity. Following the reviewers’ suggestions, we have systematically re-analysed our data, with an eye on how some deviations from linearity arise from front-end nonlinearities, and others arise from adaptive changes.

A) We acquired new data with stimulus intensity distributed over a broader range than before (Figure 1–Figure 2), which allowed us to directly visualise front-end nonlinearities in the ORN response (Figure 1). We then fit a NL model to this data, and found it could reproduce the data well, including the deviations from linearity we observed in the data (Figure 2—figure supplement 1). However, it could not account for context-sensitive decrease in response to whiffs following earlier whiffs, which we have now highlighted in our data on ORN responses to naturalistic stimuli (Figure 2, Figure 2—figure supplement 1).

B) We re-analysed our data on Weber scaling with Gaussian inputs to determine if models with an input nonlinearity could reproduce observed gain changes. We found that if the input nonlinearity is not allowed to change, this class of NL models cannot reproduce the ORN responses to increasing means (Figure 3—figure supplement 2). Instead, when we allowed the half-maximum of the input nonlinearity to vary with the mean stimulus, these “adapting” NL models reproduced the data (Figure 3—figure supplement 2). This new analysis shows that Weber-Fechner scaling with the mean stimulus originates from adaptation, and cannot arise from saturation alone, consistent with our original analysis.

C) We constructed a new model, based on the classic two-state receptor model (B Katz, 1957), where a feedback mechanism increases the free energy difference between active and inactive forms of the Or-Orco complex. This model incorporates both adaptation and saturation, and can reproduce ORN responses to naturalistic and Gaussian stimuli and reconstitutes Weber-Fechner gain scaling (Figure 8). Key differences with models proposed in earlier studies (Nagel and Wilson, 2011) are the reduced number of parameters and the architecture of the negative integral feedback, which in our case resembles more that of the classic bacterial chemotaxis system.

D) An intrinsic property of our model is that response kinetics slow down upon adaptation to increasing mean, reproducing the slowdown in LFP responses we see in the data, while preserving Weber- Fechner gain scaling. This also suggests that the temporal window of the LFP-to-spiking transformation must decrease with adaptation, to preserve the invariance of firing rate kinetics.

Validity of LN models.

*The LN models the form the core analysis tool in the paper are not validated. Several assumptions are central to the analysis. Some of these points may also originate from a lack of clarity in the paper about details of how the LN models were fit. Most importantly, it is not clear in all cases how the linear filter used to project the stimulus was computed. For example, since the standard linear filter calculation assumes symmetry of the stimulus, it is not clear how filters were computed for the naturalistic odor stimuli.*

We agree with the reviewers about the importance of properly validating LN models, and have revised the manuscript to make clear how we computed linear filters, and the assumptions we make in this linear analysis. In the revised manuscript, we largely avoid using LN models, and focus on NL and NLN models. We still use linear models for comparisons, since our definition of gain is the slope of the linear transformation of input to output. We outline below and have added to the manuscript precisely how we computed linear filters, and the caveats associated with them.

A) Filter computation. We extracted linear filters from measured Gaussian odorant stimuli and ORN responses using least-squares fitting. Given time series of input *S* and response *R*, we reshaped *S* into a matrix S^ where each row of S^ contained the stimulus up to *N* samples in the past where *N* is the length of the filter to be calculated. Using this matrix, we computed the stimulus covariance matrix C=S^T S. The linear kernel that is the best linear predictor of *R* given *S* is given by C^=C+rI However, since input signals have autocorrelation functions that can approach 0 power in some frequencies, estimated filters were often dominated by high-frequency artifacts. To remove these, we regularized *C*, using where *I* is the identity matrix and *r* is a regularization factor in units of the mean eigenvalue of *C*. We have re-written the Materials and methods to make clearer how we extracted filters. The regularization factor was optimized by cross-validation to maximize predicted variance. We now explain this method in detail in our Materials and methods section.

B) Assumptions. Linear filters can be reliably estimated even in the presence of an output nonlinearity for white Gaussian inputs (Chichilnisky, 2001). This means that in an LN system, an unbiased estimate of the true linear filter can be extracted by using white Gaussian inputs. However, since we show that our data is consistent with a saturating input nonlinearity in these ORNs, extracted filters may not be unbiased even with Gaussian stimuli. Throughout the paper, we do not assume that the filter we extract is unbiased, but rather, that it is the best linear predictor of the response given the stimulus (in the least squares sense). Filters extracted from naturalistic stimuli are clearly not unbiased since the statistics of naturalistic stimuli are far from Gaussian, but these filters are better linear predictors of the response than filters extracted from Gaussian stimuli (Figure 1—figure supplement 4). The filters extracted from naturalistic and Gaussian stimuli have similar kinetics.

C) Estimating gain without filter extraction. Gain, defined here as Δ*R* Δ*S*, is a measure of how strongly a system responds to small changes in stimulus. We used linear filters to project the stimulus to estimate gain, since ORNs have significant temporal filtering properties. We also estimated gain without extracting filters by measuring the ratio of standard deviations of the response to the stimulus, i.e., σRσS. This estimate neglects stimulus and response kinetics, and is a gross measure of the relative scales of fluctuations in the response and the stimulus. This estimate of gain agreed well with gains estimated from linear modelling, and followed the Weber-Fechner Law (Figure 3—figure supplement 1). In addition, we plotted several features in the raw data that are indicative of gain change: variable responses to similarly sized whiffs in naturalistic stimulus (Figure 2), changes in the variance, but not the mean of responses to Gaussian stimuli with increasing means (Figure 3), and relatively small changes in the variance of responses to Gaussian stimulus with switching variances (Figure 4).

*Reviewer #1:*

*This paper describes adaptation to the mean and variance of odorant stimuli, with a particular focus on adaptation to natural inputs. The question is quite interesting, and for the most part I found the conclusions of the paper well supported by the data presented. Several issues, however, limit my enthusiasm.*

*How good are LN model fits to the data?*

*The paper relies extensively on LN models to analyze adaptation. More validation of this approach is needed – e.g. how much of the response variance do LN models describe? Related to this point – are the linear filters used in analysis of responses to naturalistic inputs derived from responses to Gaussian noise inputs (this seems to be the case from the Methods)? If not, how are the filters computed? If the filters are computed for Gaussian noise inputs, are they still valid for naturalistic inputs? Similarly, in Figure 4 – are the low variance and high variance filters assumed to be the same, and is this indeed correct?*

We have added a quantification of the variance explained by all linear models (Figure 1, Figure 1—figure supplement 4, Figure 3, Figure 4—figure supplement 1).

We computed filters directly from the naturalistic inputs by using ordinary least squares fitting. (We used ordinary least squares to fit filters throughout, including to Gaussian data. In the limit of infinitely long, white stimuli, this method equals reverse-correlation; for finite stimuli, ordinary least squares gives a better fit by taking into account the exact repertoire of correlations in the presented stimulus.) These natural scene filters differed slightly from filters derived from Gaussian stimuli (Figure 1—figure supplement 4), but described more of the variance in the naturalistic data than filters derived from Gaussian stimuli. We show both filters, and compare their predictions to the response (Figure 1—figure supplement 4).

For the experiment where we changed the variance of Gaussian stimuli (Figure 4), we individually estimated filters in the low and high variance epoch. We found that these two filters were very similar. We therefore used a single filter in the analysis, computed over both high and low variance epochs. All three filters are shown in Figure 4—figure supplement 1.

LFP as a proxy for transduction.

*The LFP is used to estimate properties of odor transduction. It is not clear from the paper how good a proxy the LFP is for transduction, specifically whether it is accurate enough to support the conclusions drawn. More verification of the validity of using the LFP is needed.*

We agree with the reviewer that the LFP does not correspond exactly to the transduction current. Though it is an imperfect measure, it is a useful proxy for transduction in olfactory sensilla, and has been used and partially validated by previous literature. First, Nagel & Wilson (Nagel and Wilson, 2011) use the LFP as a proxy for transduction responses, and studied how prolonged pulses adapt LFP responses to subsequent pulses. They also showed that the LFP was unaffected by the addition of TTX, which eliminated neural spiking. Further, it is unaffected when the neuron’s partner cell in the sensillum is genetically ablated, when that partner does not sense the odorant. These two published experiments show that the LFP signal is generated by the sensing neuron, and is upstream of spiking machinery. Second, Cafaro (Cafaro, 2016) used the LFP as a proxy for transduction responses, and studied how pulses of odorant on increasing backgrounds adapted LFP responses. Finally, the two results from the LFP in our paper – Weber-Fechner gain control at the LFP level and a slowing of the LFP with increasing mean stimulus – agree with measurements of transduction current in patch-clamped ORNs in an ex-vivo preparation (Cao et al., 2016) suggesting that the effect of adaptation on the LFP we observe is consistent with the effect of adaptation observed on transduction currents in the ORN. Our revised manuscript presents more of these details to justify using the LFP as an imperfect proxy for transduction current.

Gain definition.

*The gain in Figure 2 is defined as the slope of the firing rate vs. projected stimulus curve. But doesn't that mean comparing different firing rates for different whiffs – and hence potentially confounding adaptive effects with static nonlinearities in the responses? This issue also comes up in Figure 3.*

We agree with the reviewer – this method did not distinguish between the effects of adaptation and static nonlinearities during the response to naturalistic stimuli. As mentioned earlier we now use a different method to analyse separately the contributions of the input nonlinearity and those of adaptation. For the same reason, we have removed panel Figure 3 in the old manuscript.

Note that we still define gain as Δ*S* Δ*R*, and changes in gain are defined relative to a linear model, similar to (Baccus and Meister, 2002; Rieke, 2001).

Compensation for slowing of transduction.

*The conclusion that speeding of post-transduction processes with adaptation compensate for slowing of transduction is not well developed, especially for a major conclusion of the paper. This relates to the point above about using the LFP as a proxy for transduction.*

We have expanded our figures showing this effect to better develop this result (Figure 7). We have also expanded and improved the clarity of the text corresponding to this section in the Discussion. Patch-clamp measurements from individual ORNs have confirmed that current kinetics slow down on a background (Cao et al., 2016), partially validating our use of the LFP kinetics as a proxy for transduction kinetics, and confirming part of our result on complementary kinetics of ORN response.

*Reviewer #2:*

*[…] At a big-picture level, the Introduction sets the stage well, the manuscript as a whole appears scholarly, and the claims seem well-supported by the data and analyses. A key component of this work is the impressive control over stimulus delivery, which is much harder for olfactory stimuli than for visual stimuli, and the authors clearly went to great lengths to achieve this control. Consequently, this study has a lot to recommend it, and little to dislike. My concerns are minor, focusing on the clarity of descriptions, particularly in the Materials and methods section (which is not up to the standard set in the rest of the manuscript). In its current state I suspect that readers will have considerable difficulty understanding exactly what was done at several points in the manuscript. There are some places where I question details of the results, but it seems likely that this is due to a misunderstanding of the precise form of the analysis.*

*Figure 2: one oddity is that the tangent of the f-s curve even at 0 appears strongly correlated with the ultimate size of the excursion. Naively, this would appear to violate causality, but of course it could reflect correlations in the stimulus. Nevertheless, it begs the question of how much of this analysis is really dominated by the stimulus correlations; there don't appear to be any examples of a large stimulus that were not preceded by some adaptation, and that leads to concern about the thoroughness of this particular test and analysis.*

While we cannot rule out the effect of stimulus correlations, as hypothesised by the reviewer, in determining the determining the features of this plot, its inability to distinguish between adaptation and front-end nonlinearities led us to remove this method of analysis. As mentioned above, in this new manuscript we now distinguish between the effects an input nonlinearity and adaptation have on ORN response (Figure 1–Figure 2, Figure 2—figure supplement 1). In particular, the new Figure 2takes the reviewer’s comment into account and shows how the response to whiffs varies as a function of the intensity of preceding whiffs.

*Reviewer #3:*

*This paper examines the responses of Drosophila olfactory receptor neurons to dynamic fluctuating stimuli and analyzes their responses predominantly using linear filter methods. Based on these analyses the authors argue that ORNs exhibit fast adaptation that allows them to better encode the properties of odor plumes. The experiments appear to be carefully performed and the data appear to be solid. However, I have concerns about the novelty of the approach and findings, as well as some of the analyses and interpretations.*

*Linear filter methods have been used for many years to approximate the response of a sensory neuron to a complex time-varying stimuli. A common theme of these studies is the finding that the linear approximation of the response varies as a function of stimulus statistics such as the mean and variance (e.g. Fairhall 2001, Baccus and Meister 2002). In particular, the approximate gain of the neuron generally decreases with increased stimulus variance, a phenomenon known as "gain scaling" or "variance scaling". Following on these studies several groups asked whether single neurons could exhibit this property in response to current injection, and found that it could. Subsequent modeling studies have shown that Hodgkin-Huxley models and other spiking models are capable of producing gain scaling (e.g. Lundstrom et al., 2008). Therefore, the finding that Drosophila ORNs exhibit gain scaling, and that some of this scaling arises at the level of spike generation does not seem novel or surprising.*

We agree with the reviewer that there exists a large body of work, both theoretical and experimental, that demonstrates that variance gain control is a property of spiking neurons. To our knowledge, we are the first to demonstrate that *any* neuron in the primary olfactory sensory system exhibits variance gain control – a question that has remained unanswered for long due to difficulties in stimulus control. Furthermore, our results suggest that, unlike gain control that scales with the mean stimulus, variance gain control is distributed across spiking and transduction – a novel result.

*In the Drosophila olfactory neurons studied here, several papers have examined responses to temporally modulated stimuli (e.g. Kim 2010, Nagel 2011, Martelli 2013, Cafaro, 2016, Cao, 2016). Multiple studies have found that transduction response magnitude (measured intracellularly or by LFP) is well described by a Hill equation, (that is, it rises sigmoidally with log odor concentration), as one would predict from the basic biochemistry of receptor activation (Cao, Cafaro, Nagel). Second, they have found that OR-expressing ORNs exhibit calcium-dependent adaptation that dynamically adjusts sensitivity and gain (Cao, 2016). The approach taken here confounds these two phenomena, as it assumes that responses are a linear function of concentration (here shown to be linearly related to PID voltage). Therefore, the "adaptation" reported by the authors likely contains contributions from the nonlinearity inherent in receptor activation, as well as true dynamical adaptation. This might be addressed by computing filters based on the log concentration, rather than linear concentration, and by comparing "adaptation" (estimated from projections onto linear filters) in OR- and IR expressing ORNs, as IR-expressing ORNs do not exhibit calcium-dependent adaptation, but do show activation nonlinearities.*

We agree with the reviewer that we neglected the role of front-end nonlinearities that arise from odor-receptor binding in our previous analysis and interpretation. We have thoroughly revised the manuscript taking this into account. Using models that incorporate a front-end nonlinearity (a Hill function, like the ones measured in (Cao et al., 2016)), we distinguished between features of the response that could be attributed to dynamic adaptation, and those that were consequences of a static front-end nonlinearity. For example, we found that a model with a front-end nonlinearity could reproduce responses to naturalistic stimuli well (Figure 2—figure supplement 1), suggesting that some of the deviations from linearity could arise from this input nonlinearity. However, it could not reproduce the context-dependent modulation of responses we observed in the data (Figure 2). In fact, NL model responses to whiffs occurring after preceding whiffs were *larger* than to those occurring in relative isolation (Figure 2—figure supplement 1) – a trend opposite to what we observe in the data. This suggests that this feature of the response arises from adaptation, not saturation. We thank the reviewer for bringing up this important distinction, and have re-analysed and re-written our manuscript to make this difference clear.

*An issue in the paper is what is meant by "adaptation." The authors seem to argue that adaptation is any change in the linear approximation to the response. For example, in the Introduction they state that they are focusing on phenomenology, which they call "a prerequisite to understanding the mechanisms that implement them." In the Discussion, they say that "'adaptation state is a construction to understand nonlinearities in that response." However, many other parts of the paper imply that these "adaptations" are actively produced by the cell in order to optimize coding. For example, in the title "Olfactory receptor neurons use fast dynamic gain control", Results "small isolated whiffs were amplified…suggesting that ORNs adjusted their gain dynamically", "suggesting that ORNs actively changed their gain." The question of whether certain coding phenomena arise from active adaptation processes, or from unavoidable nonlinearities of receptor-ligand interactions seems germane to the framing of the paper, and the question of whether the phenomena they describe are specializations of insect neurons for encoding odor plumes or general properties of receptor-activated neurons.*

The term “adaptation” has been used to refer to a wide variety of phenomenon in ORN response kinetics: (1) the decrease in response following an initial transient on a step change in stimulus (Cafaro, 2016; Cao et al., 2016); (2) the decrease in responses to brief test pulses following previous, identical, pulses (Cao et al., 2016); (3) the decrease in response to brief test pulses following longer conditioning pulses (Nagel and Wilson, 2011); and (4) the decrease in response to brief test pulses presented on a background, *vs.* those presented without a background (Cafaro, 2016; Cao et al., 2016; Martelli et al., 2013). We agree with the reviewer that the division of these phenomena into those that arise from consequences of input nonlinearities and those that arise from active change in system parameters is an important question. In our revised manuscript, we have generated figures and fit models to make those distinctions. We find that in our naturalistic stimuli, gain changes both due to saturation of instantaneous front-end nonlinearities (seen in Figure 1), and also in response to the stimulus in the recent past (Figure 2). Using models with an input nonlinearity, we show that gain changes due to stimulus in the recent past cannot be reproduced by NL- style models (Figure 2—figure supplement 1), but can be reproduced if the input nonlinearity is allowed to adapt (Figure 8). The adaptive two-state model that we now use in the paper makes more precise contributions from “active” and “passive”: an interesting aspect of the model is how the “active” adaptation of the free energy of the Or-Orco complex not only affects the properties of the “passive” dose response, but also the kinetics of switching between active an inactive state, i.e. the “active” part of the response to whiffs.

*A second issue is the characterization of phenomenology versus mechanism. Although the authors claim to focus on phenomenology, a significant part of the paper is devoted to the distinguishing phenomena arising at the level of transduction/LFP, and those arising at the level of spikes- an attempt to place phenomenology in a mechanistic context. Given that multiple stages have been experimentally identified in the transduction process (Cao et al. 2016), and that many models of spike generation are available, it does not seem fair to call this phenomenological characterization a "necessary prerequisite" to mechanistic understanding.*

We believe that phenomenological descriptions are important for determining which mechanistic details are important. However, we agree that our manuscript also makes some effort to constrain potential mechanisms. Therefore, we have toned down this sentence and made sure to acknowledge that papers in the literature are already dissecting mechanistic details of these neurons.

*Other comments:*

*The finding that ORN spikes seem to counteract some of the slowing of transduction responses is interesting and novel but not well-developed.*

We agree that this is an interesting discovery. In the revised manuscript, we developed this aspect further. We now devote a full figure to this discovery (Figure 7). We have also developed a model based on the classic two-state receptor model (B Katz, 1957; Nagel and Wilson, 2011) but with an adaptive architecture that resembles that of bacterial chemotaxis, and found that this model naturally decreases its response kinetics with adaptation to the mean stimulus. We have used this to reproduce the slowdown in LFP responses we observed with a single set of parameters (Figure 8). Thus, we have developed this aspect of our manuscript significantly in revision.

*I am concerned about the linear filter method applied to responses to sparse stimuli in Figure 2. As stated in the Materials and methods, linear filters can be reliably estimated in the presence of an output nonlinearity if the stimuli are Gaussian- however, this is generally not true for stimuli that are strongly skewed, such as the one used here.*

[Note: former Figure 2is now Figure 1]. We computed filters directly from the naturalistic inputs by using ordinary least squares fitting. (We used ordinary least squares to fit filters throughout, including to Gaussian data. In the limit of infinitely long, white stimuli, this method equals reverse-correlation; for finite stimuli, ordinary least squares give a better fit by taking into account the exact repertoire of correlations in the presented stimulus.) These natural stimulus filters differed slightly from filters derived from Gaussian stimuli (Figure 1—figure supplement 4) but described more of the variance in the naturalistic data than filters derived from Gaussian stimuli. We show both filters, and compare their predictions to the response (Figure 1—figure supplement 4). We have added a quantification of the variance explained by all linear models (Figure 1, Figure 1—figure supplement 4, Figure 3, Figure 4—figure supplement 1).

In addition, we have expanded our analysis of responses to naturalistic stimulus to include methods that do not rely on computing a linear filter. By comparing responses to whiffs of similar amplitude, we show that LFP and firing rate responses to similar-sized whiffs are smaller when those whiffs occur after other whiffs (Figure 2).

*In general the authors do not report the goodness of fit of their linear model, making it difficult to interpret whether the observed changes in gain (where gain is the slope of the projected versus real response) are significant.*

We have added a quantification of the variance explained by all linear models (Figure 1, Figure 1—figure supplement 4, Figure 3, Figure 4—figure supplement 1).

In addition, we also estimated gain without extracting filters by the ratio of standard deviations of the response to the stimulus, i.e., KD. This estimate neglects stimulus and response kinetics, and is a measure of the relative scales of fluctuations in the response and the stimulus. This estimate of gain agreed well with gains estimated from linear modelling, and followed the Weber-Fechner law (Figure 3—figure supplement 1). In addition, we plotted features in the raw data that are indicative of gain change: variable responses to similarly sized whiffs in naturalistic stimulus (Figure 2), changes in the variance, but not the mean of responses to Gaussian stimuli with increasing means (Figure 3), and relatively small changes in the variance of responses to Gaussian stimulus with switching variances (Figure 4).

*The authors state: "It remains unclear whether the ORN's ultimate output- the firing rate- follows the same Weber-Fechner scaling." However, Cafaro, 2016 does address this question for ORN spikes, as well as PN membrane potential and spikes.*

Cafaro (Cafaro, 2016) used pulses of odorant on backgrounds to quantify how the magnitude of responses to these pulses varied with odor background. Cafaro found evidence for Weber-Fechner gain scaling at LFPs and PNs, but not at firing rates (emphasis added):

If pulse adaptation follows Weber’s Law, the contrast–response functions at different background intensities will overlay each other (i.e., the response will depend only on the contrast not on the background intensity). The contrast–response curves for the ORN LFP appear to overlap at higher background odor intensities (>10 ^7^; Figure 6), suggesting that olfactory transduction may approach Weber’s law at higher background intensities. Much of the contrast– response curve overlap is lost in the ORN spike rate because of pulse sensitization during ORN spike generation (Figure 6). Owing to the strong pulse adaptation in the glomerular transform, the contrast–response curves overlap in the PN synaptic potential (Figure 6) and the PN spiking response (Figure 6).

In addition, since Cafaro compares neuron responses to dilution in liquid phase (x-axis in Figure 6 is liquid phase dilution of odorant), not instantaneous gas phase, and since gain at any stage is not directly estimated, it is unclear what the relationship between ORN gain and mean stimulus is, and if it varies with 1/*S* (the mean background). In our paper, we use controlled Gaussian odorant stimuli and measure their instantaneous gas phase concentration to estimate gain at three stages (i) stimulus to LFP, (ii) LFP to firing rate, and (iii) stimulus to firing rate. We found that gain at the first stage varied with the Weber- Fechner Law (Figure 5), consistent with (Cafaro, 2016; Cao et al., 2016). We found that gain at the second stage was invariant to stimulus background (Figure 5), which meant that gain at the third stage also varied with the Weber-Fechner Law (Figure 3).

*Introduction, last paragraph: It is unclear at this point in the text why the authors choose to focus on a history of 300ms. This should be better motivated.*

In the revised manuscript, this section no longer exists. In the old manuscript, we chose this duration for illustrative purposes because it showed a large effect, but also investigated many different history lengths. Figure 2 in the old manuscript repeats the analysis we illustrated for 300 ms for several timescales from 10 ms to 10 s.

In Figure 2of our revised manuscript, we used a history of 300 ms to compute the mean stimulus history over for illustrative purposes (Figure 2). To generalise, we parameterised the stimulus time series without explicitly picking a particular timescale (Figure 2).

*Subsection “Fast gain control maintains timing information of naturalistic odor signals”, third paragraph. It would be helpful to at least briefly mention how lags were computed in the Results.*

We have modified the figure (now Figure 7) showing how we compute lags through cross correlation functions. We show cross-correlation functions from the stimulus to the LFP (Figure 7), and from the stimulus to the firing rate (Figure 7), for large and small mean stimulus, showing that cross-correlation functions for the stimulus-LFP shift significantly with mean stimulus background, while cross-correlation functions for the stimulus-firing rate do not. We have also added another panel (Figure 7) showing that these differences in cross-correlation functions in the LFP exist despite similar stimulus auto-correlation functions between low and high stimuli.

*Subsection “Variance gain control in olfaction”. No potential mechanisms for gain scaling of transduction responses are given.*

In our revised manuscript, we have developed a model of receptor binding and activation based on the classic two-state receptor model (B Katz, 1957). An intrinsic property of this model is that response kinetics slow down upon adaptation to increasing mean, reproducing the slowdown in LFP responses we see in the data, while preserving Weber-Fechner gain scaling. This model is similar to the model proposed to reproduce LFP responses in (Nagel and Wilson, 2011). While in our earlier manuscript we had to modify our model to slow down reaction rates in order to reproduce the slowdown in LFP kinetics, we found that our current model naturally slows down response rates with adaptation to the mean stimulus, while preserving Weber-Fechner gain control (Figure 8). The mechanism of slowdown in response is due to the intrinsic structure of the model and the feedback of receptor activity onto activation and inactivation rates (see Materials and methods).

*Subsection “Fast gain control could aid in naturalistic odor detection”. The authors state that paradigms that employ conditioning and probe stimuli cannot easily quantify the dynamics of gain change. This does not seem accurate. These paradigms explicitly allow the experimenter to distinguish the stimuli used to generate adaptation from those used to test it. Small combinations of pulses are used to examine the dynamics of gain change in transduction in Cao et al. 2016.*

We agree with the reviewer. Our sentence was inaccurate – dynamic models can be fit to many stimulus paradigms, as noted. We have removed this sentence.

*Subsection “Stimulus measurement”. It would be helpful to include a diagram of the odor delivery apparatus along with calibration data.*

We have included a diagram of the odor delivery apparatus with the calibration data (Figure 1—figure supplement 1).

[Editors' note: the author responses to the re-review follow.]

*The reviewers have discussed the reviews with one another all agreed that the revisions have improved the paper substantially. There are a few outstanding issues outlined in the reviews below.*

*Reviewer #2:*

*[…] Overall the manuscript is much improved. I think it makes a substantial contribution to the literature. I do have some remaining issues with the manuscript that I think should be resolved prior to publication.*

1) The Hill function as a source of nonlinearity.

*The reviewers previously asked whether many of the nonlinearities described in the data could arise from receptor-ligand interactions, which should give rise to a nonlinear relationship between odor concentration and transduction. The authors have re-analyzed their data and found that responses to natural stimuli are well-described by such a relationship (Figure 1), although context-dependent deviations from this relationship remain (Figure 2). However, it seems that this understanding is not entirely integrated into the text. For example, the title of the first section is "ORN responses to naturalistic odorant stimuli show deviations from linearity that arise from adaptation and saturation" which implies that saturation is the only nonlinearity (not the logarithm implied by the Hill function). Much time is spent on showing that responses deviate from linearity, and then receptor-ligand interactions are suggested as an alternative to output nonlinearities, which seems odd, given that receptor-ligand nonlinearities are an obligate feature of a system composed of odorant receptors that bind and are activated by odorants. A finding that odor concentration is encoded linearly by such a system would be much more surprising.*

We agree with the reviewer that a Hill function imposes nonlinearities that go beyond mere saturation of the response, and have revised the title of the first section to reflect this.

*Along these lines, in the Introduction, the authors state that "ORNs employed front-end nonlinearities" which makes it sound like an active choice on the part of ORNs. I think a fairer statement of the findings would be that a front-end nonlinearity, combined with dynamic adaptation that shifts the midpoint of this nonlinearity, can account for ORN responses to a variety of dynamic stimuli.*

We have replaced “employed” with “exhibited” to remove any intent from the statement.

2) Explanation of Weber-Fechner scaling.

*Understanding the origin of Weber-Fechner scaling is a major goal of the paper. Along the lines above I would suggest that Figure 3—figure supplement 2 should be made part of the main text. This shows that the expected front-end nonlinearity, combined with adaptation that shifts the activation curve to the right, is sufficient to account for ORN responses to stimuli with different means. As such a rightward shift with adaptation has been previously reported (Kaissling, 1987, Nagel and Wilson, 2011), this makes for an elegant explanation of the observed responses.*

*The authors state that the receptor model alone cannot account for Weber-Fechner scaling (response to reviewers 1B). It is true that the receptor model alone does not produce responses that change gain and have similar mean, as in the data. However, it does seem that the receptor model alone (Figure 3—figure supplement 2, panel c) shows a decrease in gain with increased mean. If the output of this simple model were plotted as in Figure 3 what would it look like? Does the change in gain arise from the dynamic adaptation (the shift in half-max) or from the nonlinearity?*

We thank the reviewer for this suggestion. We have included a new panel to Figure 3—figure supplement 2 (panel G). In it, we compare the steady state gains when KD is fixed and when it is allowed to vary. We show that when KD is fixed, the gain varies with the mean stimulus as a power law with exponent –2, inconsistent with the Weber-Fechner Law. When KD varies with the mean stimulus, the gain varies with the mean stimulus as a power law with exponent –1, in agreement with the Weber-Fechner Law. Furthermore, when KD is fixed, the mean responses increase monotonically with the mean stimulus. When KD is allowed to vary, the mean response is constant (consistent with our data). These two features suggest that our data is consistent with adaptation and not merely a consequence of an input nonlinearity.

3) Receptor-based model.

*The authors present a new model, based on a 2-state receptor model, to account for the observed adaptation. As I understand it, their model makes two assumptions: (1) receptor activation/inactivation are the rate limiting steps, and (2) receptor activity feeds back onto these rates, slowing both. The model thus makes specific predictions about what biophysical steps give rise to adaptation.*

*Given the findings in Figure 3—figure supplement 2, it seems like the main features necessary to explain the results are (1) steady-state output rises as S/S+K, (2) adaptation shifts K to the right, (3) adaptation decreases transition rates, leading to slower responses. The authors should clarify which aspects of their model are essential for reproducing the key features of the data.*

Thank you for this suggestion. We have added the following to our discussion of the model properties and results: “…the steady state activity in our model depends nonlinearly on the stimulus, reproducing the effects of saturation in responses to naturalistic stimuli. Adaptation shifts the effective half-maximum of the input nonlinearity to the right, recapitulating Weber-Fechner gain control; and adaptation decreases transition rates from active to inactive receptor complexes, reproducing slowing LFP responses with adaptation.”

4) Use of the LFP as a proxy for transduction.

*It is true that it has become somewhat standard in the literature to do this but the LFP should be interpreted with a bit of caution. In Nagel and Wilson 2011, palp LFPs were used because LFPs in this structure were found to more closely reflect activity in a single sensillum (presumably because of the greater spacing between sensilla). A brief caveat noting that LFPs can contain a contribution from nearby sensilla would be welcome.*

We agree with the reviewer; the LFP is not directly a measure of transduction. We have expanded our section detailing the caveats of using the LFP to refer to the fact that LFP could reflect contributions from nearby sensilla.